# Tackling recalcitrant *Pseudomonas aeruginosa* infections in critical illness via anti-virulence monotherapy

Vijay K. Singh [1,2,3,14], Marianna Almpani[1,2,3,14], Damien Maura[1,2,3,11,14], Tomoe Kitao [1,2,3,12], Livia Ferrari[4], Stefano Fontana[5], Gabriella Bergamini[4], Elisa Calcaterra[4], Chiara Pignaffo[5], Michele Negri[6], Thays de Oliveira Pereira[7], Frances Skinner [8], Manos Gkikas [8], Danielle Andreotti[9], Antonio Felici[10,13], Eric Déziel[7], Francois Lépine[7] & Laurence G. Rahme [1,2,3] ✉

Intestinal barrier derangement allows intestinal bacteria and their products to translocate to the systemic circulation. *Pseudomonas aeruginosa* (*PA*) super-imposed infection in critically ill patients increases gut permeability and leads to gut-driven sepsis. *PA* infections are challenging due to multi-drug resistance (MDR), biofilms, and/or antibiotic tolerance. Inhibition of the quorum-sensing transcriptional regulator MvfR(PqsR) is a desirable anti-*PA* anti-virulence strategy as MvfR controls multiple acute and chronic virulence functions. Here we show that MvfR promotes intestinal permeability and report potent anti-MvfR compounds, the N-Aryl Malonamides (NAMs), resulting from extensive structure-activity-relationship studies and thorough assessment of the inhibition of MvfR-controlled virulence functions. This class of anti-virulence non-native ligand-based agents has a half-maximal inhibitory concentration in the nanomolar range and strong target engagement. Using a NAM lead in mono-therapy protects murine intestinal barrier function, abolishes MvfR-regulated small molecules, ameliorates bacterial dissemination, and lowers inflammatory cytokines. This study demonstrates the importance of MvfR in *PA*-driven intestinal permeability. It underscores the utility of anti-MvfR agents in maintaining gut mucosal integrity, which should be part of any successful strategy to prevent/treat *PA* infections and associated gut-derived sepsis in critical illness settings. NAMs provide for the development of crucial pre-ventive/therapeutic monotherapy options against untreatable MDR *PA* infections.

The recalcitrant ESKAPE pathogen *Pseudomonas aeruginosa* presents a serious threat to critically ill and immunocompromised patients[1–3]. The most prevalent among the frequently colonized by *P. aeruginosa* body sites are the respiratory system, urinary tract, skin, and gastrointestinal tract[4–7]. This opportunistic pathogen's intestinal colonization, in particular, is associated with elevated mortality rates of patients in intensive care units (ICU), and its significance as a cause of mortality in critically ill patients has been demonstrated in randomized prospective studies[8,9]. Besides being an important reservoir for this pathogen, the gastrointestinal tract can be a significant source of systemic sepsis and death among critically ill patients[10,11] who inherently have defective intestinal integrity secondary to their critical clinical condition[12,13].

A list of author affiliations appears at the end of the paper. ✉e-mail: rahme@molbio.mgh.harvard.edu

There is an exponentially growing body of evidence that the composition of the gut microbial flora and alterations of the commensal bacterial populations following injuries, infections, and critical illness, strongly influence our metabolic, endocrine, immune, peripheral, and central nervous systems. It is now clear that maintaining gut mucosal integrity must be part of any successful strategy to prevent/ treat infections and the gut-derived sepsis syndrome seen in critical illness settings. Derangement of the intestinal barrier subsequently allows intestinal bacteria and their products to translocate to the systemic circulation. A significant number of lung infections have been reported to arise due to direct contamination of the airways by the gastrointestinal flora or by hematogenous dissemination from the intestine to the lung parenchyma[10,11,14,15]. At the same time, the crucial role of circulating microbes originating from the gut has long been recognized as a critical player in the development of multiple organ failure (MOF) in critically ill patients[16]. When added to the critical illness, *P. aeruginosa* infections occur, and there is an exacerbation of the intestinal barrier dysfunction with even more devastating results. Indeed, *P. aeruginosa* virulence factors have been shown to further promote increased intestinal permeability[17].

Infections with *P. aeruginosa* are challenging to eradicate due to this pathogen's high antibiotic resistance[18,19]. Moreover, attempts to eradicate *P. aeruginosa* infections can fail when traditional antibiotics leave unharmed the subpopulation of bacterial cells that are refractory to antibiotics[1], which along with biofilms are ultimately responsible for chronic and persistent/relapsing infections[20,21]. Therefore, in the post-antibiotic era, the development and implementation of new antimicrobial strategies would allow us to tackle multi-drug resistant (MDR) infections effectively, and the formation of antibiotic tolerant persister (AT/P) cells is imperative. One attractive and intensely investigated anti-microbial approach is quorum sensing (QS) inhibition[22–24], a cell-cell communication signaling mechanism employed by bacteria to efficiently coordinate their behaviors, many of which are virulence-related. Bacteria, including *P. aeruginosa*, release low molecular weight molecules as chemical signals capable of concomitantly mediating the transcription of virulence genes[25] and

modulating host immune responses[26–28]. QS inhibition (QSI) is neither bactericidal nor bacteriostatic. Therefore, the principle behind the QSI approach to treat severe infections is to disarm the virulent bacteria, rendering them less pathogenic for the host while simultaneously avoiding the strong selective pressure that antibiotic killing or antibiotic-mediated growth arrest entails.

The QS transcriptional regulator MvfR (multiple virulence factor regulator, also known as PqsR)[29,30] is one of the three interconnected *P. aeruginosa* QS regulators that govern many virulence functions in this pathogen[31–36] (Fig. 1). MvfR plays a central role in the *P. aeruginosa* QS interplay due to its direct control of the other QS regulators, LasR and RhlR. In addition, MvfR controls the synthesis of ~60 distinct low-molecular-weight compounds via the transcriptional regulation of the *pqsABCDE* operon[36]. The production of these small signaling molecules is responsible for the difficulty of eradicating acute, chronic, and persistent/relapsing (ACPR) infections, including acute and chronic pneumonia, relapsing and chronic wound and ear infections, as well as medical device-related infections[37]. We have shown that loss of the MvfR function completely abolishes the production of these molecules and several redox-generating molecules[34,38]. These include reduced production of pyocyanin, a redox-active molecule that causes oxidative stress in host cells and dysregulates the host immune mechanisms[28,39,40] and, importantly, 2-AA (2-aminoacetophenone) that promotes LasR mutations, AT/P cell formation, and host chromatin modifications, impacting histone acetylation. As shown in Fig. 1, all these effects lead to persistent infections[28,31,32].

The ability of MvfR to control acute and chronic bacterial functions, and notably, unlike LasR, the other QS regulator, no clinical isolates from patients have been reported to date to have frequent mutations in MvfR, making it a highly desirable target for drug discovery and underscoring its importance in *P. aeruginosa* pathogenesis[41–43].

In this study, we focused on one of the unexplored chemical families we identified from our original whole-cell-based High-Throughput Screen (HTS) represented by the non-ligand-based compound M17[22] and built upon the harnessed knowledge on the MvfR

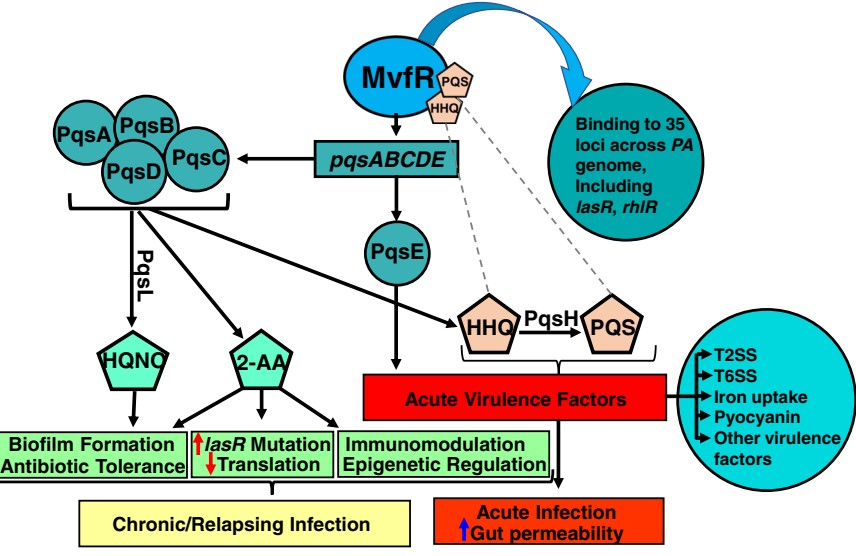

**Fig. 1 | Current view of the *P. aeruginosa* MvfR QS system impact on acute and chronic functions.** MvfR (PqsR), in the presence of its ligands/inducers PQS or HHQ, binds and activates the transcription of the *pqs* operon, whose encoded proteins catalyze the biosynthesis of ~60 low molecular weight molecules, including HHQ, HQNO, and 2-AA. HHQ in turn is converted to PQS via PqsH. HQNO promotes pro-acute and pro-persistent phenotypes, while HHQ and PQS promote acute phenotypes. 2-AA's immunomodulatory action and epigenetic regulation and its ability to promote the accumulation of *lasR* mutants and

formation of AT/P cells that survive antibiotic killing contribute to persistent infections. Moreover, MvfR impacts the production of several virulence factors, including pyocyanin, and binds and directly regulates the expression of 35 loci across the *P. aeruginosa* genome, including major regulators and virulence factors, such as the QS regulators LasR and RhlR, and genes involved in protein secretion, translation, and response to oxidative stress[34]. Agents that bind and inhibit MvfR function are a successful strategy to control the multiple virulence functions under MvfR control.

function, its regulation, and inhibition[12,22,23,32,44–47]. We demonstrate the importance of MvfR in intestinal permeability and open new avenues for preclinical development of anti-MvfR agents based on the family of highly efficient chemical series, namely N-Aryl Malonamides (NAMs) that are compatible with in vivo use.

## Results

### Design and structure-activity relationship (SAR) of non-ligand-based anti-*MvfR* agents

Our original whole-cell-based HTS of a chemical library of 284,256 molecules identified several chemical families of compounds with a potent anti-MvfR activity that were structurally different from the MvfR positive regulatory ligands 4-hydroxy-2-heptylquinoline (HHQ) and 3,4-dihydroxy-2-heptylquinoline (PQS)[22,33]. Previously we focused on the family of compounds containing a benzothiazole moiety and a variously substituted aromatic amide group[22], such as M64 (Supplementary Fig. 1a). While many of these compounds were quite active in inhibiting MvfR function, they suffered from low solubility and presented chemical liabilities that complicated their utility in an in vivo setting. Notably, the presence of the sulfur atom seemed problematic as it is susceptible in vivo to metabolism.

Here we investigated the anti-MvfR potential of another previously unexplored chemical family identified from our original HTS. Starting with compound M17[22] and following many rounds of structure-activity relationship (SAR) studies, a total of 84 compounds were generated and tested (Supplementary Tables 1–7) to identify a chemical series, N-Aryl Malonamides (NAMs), as potent MvfR inhibitors non-based on the structure of the MvfR-native ligands. Given that MvfR regulates the transcription of the *pqsABCDE* operon and the production of pyocyanin, as a first step of the compounds' activity screening and prioritization, we determined their efficacy in impacting pyocyanin production and the transcription of the MvfR-regulated *pqs* operon using a *pqs*-GFP reporter construct (Supplementary Tables 1–7)[48].

M17 (Supplementary Table 1 and Supplementary Fig. 1b) shares the substituted N-Aryl amide moiety of our previously identified low soluble chemical family of anti-MvfR compounds[22]. Compared to the potent compound M64 (Supplementary Fig. 1a), M17 has a 2-methyl, 4-Fluoro anilines in place of the 2-tio-benzimidazole motif and a 4-cyano instead of 4-phenoxy on the N-Aryl amide side. A series of M17 analogs (N 2-(Arylamino)-N-arylacetamide) was generated by replacing the various substituents on both aryl-rings (Supplementary Tables 1 and 2) without obtaining compounds significantly more active than M17. Varying the nature of the linker and the chain length between the two substituted Aryl rings (Supplementary Table 3), no particularly relevant results were observed except for D24, where the central glycinamide has been formally attached with the opposite orientation. D24 showed a similar anti-MvfR profile to D16, D28, and D33 (Supplementary Table 2), suggesting that the linker (the glycinamide) can be placed between Aryl rings in either way. That prompted us to design D36, characterized by a malonamide as a central motif, and this compound was selected for further optimization (Supplementary Tables 4–7).

The most active malonamide derivatives generally share two substituents at the para position of each of the phenyl rings. These substituents usually are electron-withdrawing and/or lipophilic groups. The presence of groups such as cyano, chlorine, bromine, trifluoromethyl, nitro, or iodine on one or both aromatic rings increases the compounds' anti-MvfR activity. Substituting one phenyl ring with the lipophilic para phenoxy group led to a series of very active compounds such as D57 (Table S4), D67, D68, D88, and D100 (Supplementary Tables 5, 6 and Fig. 2). The presence of two substituents between the two carbonyls resulted in a loss of potency for compounds D87, D94, and D97, suggesting the need to have at least one hydrogen available in the position as observed for the symmetric

compounds D43 with D80. A single fluoride in that position, as in compound D96, was partially tolerated. However, when combined with a more lipophilic group (Phenoxy) by replacing one of the 4-cyano group presents on the aryl gave rise to compound D88, which retained the potency and surprisingly increased the solubility (Supplementary Table 8) despite having the lipophilic phenoxy substituent generally responsible of reducing solubility.

Our results show that of the 84 compounds tested, 18 compounds, D36, D41, D42, D43, D51, D56, D57, D58, D61, D62, D63, D69, D71, D95, D77, D80, D88, and D100, reduced the transcription from the *pqsA* gene expression inhibiting its transcription by ≥ 90% (Supplementary Tables 2–7). Similarly, compounds D36, D41, D42, D43, D51, D57, D58, D60, D61, D62, D63, D69, D92, D95, D77, D80, D88, and D100 inhibited pyocyanin production also by ≥90% (Supplementary Tables 1–7). None of the compounds tested affected the growth of any *P. aeruginosa* clinical isolates used in these studies (Supplementary Figs. 2 and 3b), which is characteristic of anti-virulence compounds.

To functionally validate these results, we subsequently tested the efficacy of these compounds in inhibiting the synthesis of the *pqs* operon catalyzed small excreted 4-hydroxy-2-alkylquinolines (HAQs) molecules, including its positive regulatory ligands HHQ and PQS, the biofilm-related molecule 4-hydroxy-2-heptylquinoline *N*-oxide (HQNO); and the non-HAQs, 2,4-dihydroxyquinoline (DHQ) and 2-aminoacetophenone (2-AA), in the presence of the compounds[32,33,36,49]. Table S1 shows that among the tested compounds (at 50 μM), D41, D42, D43, D51, D61, D62, D63, D69, D77, D80, D88, and D95, had the most robust inhibitory profile in agreement with their inhibitory efficacy against pyocyanin production and *pqsA* gene expression (Supplementary Tables 1–7). Specifically, HHQ, PQS, and HQNO production inhibition ranged between 89–99%, 76–97%, and 40–92%, respectively. In addition, inhibition of 2-AA and DHQ production ranged between 82–95% and 82–96%, respectively.

Based on all compounds' inhibition profiles in all the functions tested, we selected for advancement the 10 most potent compounds D41, D42, D43, D57, D69, D77, D80, D88, D95, and D100 (Supplementary Tables 1–7 and Fig. 2). These compounds do not contain a chemical group or atom such as a sulfur atom or a free amino group that could easily undergo oxidation in vivo.

### NAMs inhibit MvfR-regulated virulence functions in the micromolar range

To determine the compound dose-efficacy relationship, we further measured the concentration at which the 10 aforementioned compounds exert their 50% inhibitory effect on pyocyanin production assay and *pqsA* gene expression. Figure 2 shows the dose-dependent inhibition measured in the PA14 strain for all 10 compounds for both assays. The range of the $IC_{50}$ values for the 10 compounds in the pyocyanin production was 0.15–4.07 μM. More than half of the compounds had an $IC50 ≤ 1$ μM (D57: 0.24 μM; D69: 0.96 μM; D88: 0.53 μM; D95: 0.90 μM; D100: 0.38 μM) with D57 having the lowest (0.24 μM) $IC_{50}$ for pyocyanin production and D77 having the highest of all (4.07 μM) (Fig. 2). For the *pqsA*-GFP expression assay, the range of the $IC_{50}$ values was 0.43–10.6 μM, with the vast majority of the compounds having an $IC_{50}$ around or below 1 μM (D41: 1.09 μM; D42: 0.92 μM; D43: 0.97 μM; D57: 0.43 μM; D69: 1.7 μM; D88: 1.31 μM; D95: 1.09 μM; D100: 0.38 μM), and the lowest $IC_{50}$ value being observed following incubation with D57 (Fig. 2).

Sixteen MDR *P. aeruginosa* blood or wound isolates (Supplementary Table 9) were used to cross-validate the efficacy of all 10 NAMs in inhibiting pyocyanin production. As shown in Supplementary Fig. 3a, all except D80 are highly efficacious against all multidrug-resistant *P. aeruginosa* clinical isolates tested. At a concentration of 10 μM NAMs, most compounds substantially reduced pyocyanin production in these MDR *PA* clinical isolates (Supplementary Fig. 3a).

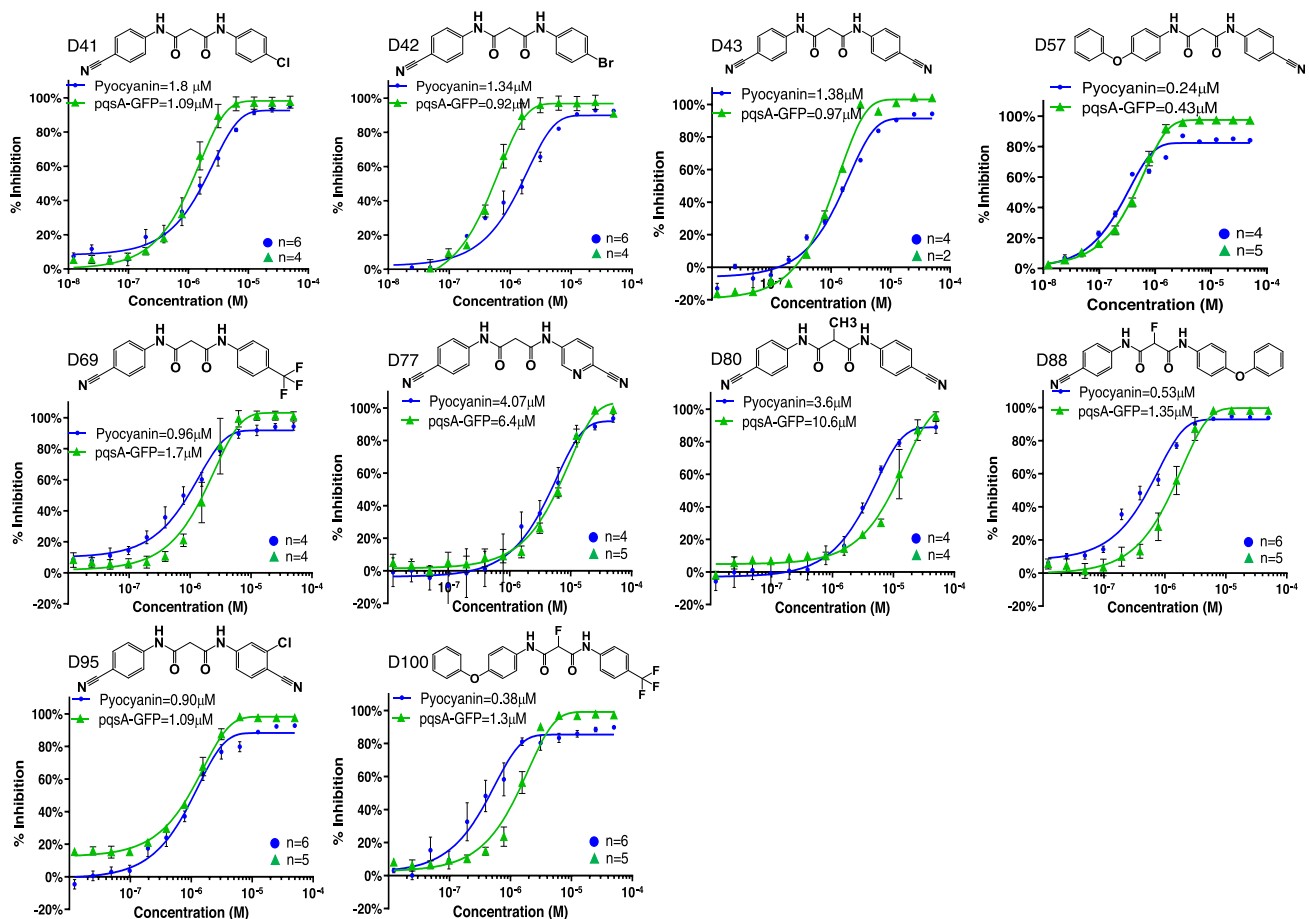

**Fig. 2 | Dose-dependent inhibition of pyocyanin production and *PqsA-GFP* expression.** $IC_{50}$ determination was performed in the presence and absence of each of the 10 NAM compounds at 13 different concentrations ranging from 0.01–50 μM. Graphs are showing $IC_{50}$ values for pyocyanin (blue) and *PqsA-GFP* (green) for D41, D42, D43, D57, D69, D77, D80, D88, D95, D100. $IC_{50}$ curves were plotted as a percentage of pyocyanin production and *pqsA-GFP* expression of the indicated compounds. The percentage of compound inhibition was calculated by comparing PA14 cells grown in the presence of the vehicle control. The $IC_{50}$ values for each compound were calculated using the GraphPad Prism 9.2.0 software. Data represent the mean value of biological replicates, and the number (*n*) of replicates is indicated on the graph. The error bars denote ±SEM. *n* = number of biological replicates; NAM = N-Aryl Malonamide.

## NAMs prevent the formation of *P. aeruginosa* AT/P cells and attenuate the initiation of biofilm formation

The MvfR-regulated signaling molecule 2-AA promotes the formation of the AT/P cells[31] implicated in the failure of antibiotic treatments in clinics. AT/P cells, a subpopulation of bacterial cells that survive lethal concentrations of antibiotics, can lead to persistent bacterial infections that can be the source of latent, chronic, or relapsing infections. Therefore, we further tested the efficacy of the 10 advanced NAMs against the formation of AT/P cells and compared their efficacy to the wild-type strain PA14 and isogenic mutant *mvfR*. Figure 3a shows that all 10 compounds significantly reduced the formation of AT/P cells compared to the wild-type strain PA14, agreeing with their efficacy to decrease the levels of 2-AA synthesis (Supplementary Tables 1–7). Most of them provided a significant reduction similar to that observed with the *mvfR* mutant. Specifically, the 5 compounds D41, D42, D57, D69, and D88 inhibit AT/P cells formation by ≥76% (76%–88%) and the rest of D43, D77, D80, D95, and D100 by ≥51% (56%–70%).

Moreover, Fig. 3b shows that initiation of biofilm formation was also significantly reduced by all the tested compounds after 24 h of growth in the presence of 10 μM of the respective compounds compared to the PA14 control strain. Specifically, 8 out of the 10 compounds, namely D41, D42, D69, D77, D80, D88, D95, and D100, showed an inhibition rate between 80–45%, while D43, D57, D80, and D95 exerted an inhibition rate of 40–30%. The highest inhibition was observed by compound D100 followed by compounds D77 and D88.

## Assessments of NAMs targeting MvfR

To further determine NAMs specificity for MvfR and exclude their potential binding to any of the Pqs operon enzymes, we tested their ability to inhibit the production of HAQs, 2-AA, and DHQ utilizing the isogenic *mvfR* mutant constitutively expressing the *pqsABCDE* operon, which leads to theproduction of these molecules independently of MvfR. Figure 4a shows that the levels of these molecules are similar between all the compound-treatment groups and the vehicle-treatment control, confirming that our selected NAMs do not confer any MvfR-independent inhibition of the HAQs, 2-AA, and DHQ production by interfering with the enzymes catalyzing their synthesis. Moreover, to further assess the absence of an off-target effect with the selected NAMs for the MvfR-related phenotypes tested, we assessed the formation of AT/P cells and biofilm using the PA14 isogenic *mvfR* mutant. Figure 4b, c shows that none of the MvfR NAM inhibitors tested exhibited an off-target effect since the formation of AT/P cells and biofilm profile were similar to that of the *mvfR* mutant alone or in combination with all of the tested NAMs.

Surface plasmon resonance (SPR) analysis was also performed to determine whether the advanced compounds bind MvfR and assess their binding affinity. Supplementary Fig. 4 shows that these compounds indeed bind MvfR with high affinity having a $K_D$ value ranging between 0.24–1.25 μM.

Taken together, these findings show that the advanced NAMs inhibit MvfR-regulated functions by targeting MvfR.

## Solubility assessment prioritizes D88 as the lead compound

A compound's low solubility can be a bottleneck in drug development. Solubility measurements using High-Performance Liquid Chromatography (HPLC) show that the 10 compounds' solubility ranged between 2 µM and 490 µM (Supplementary Table 8). Although several of these 10 compounds performed consistently well in all the assays mentioned above, compound D88 displayed by far the highest solubility (490 µM) of all the tested compounds. It was one of the most effective inhibitors in all assays performed. Notably, the increase in solubility observed with D88 was not observed with D100, likely due to the extra lipophilicity added by the $CF_3$ group replacing the Cyano. Replacing one or both aromatic groups with a pyridine, as in D77, shows low solubility (Supplementary Tables 7 and 8). Thus, we focused on D88 for further assessments prior to using it in vivo studies.

## Additional studies support the prioritization of D88

First, we assess the potency of this NAM in the inhibition of PqsABCD products by determining the $IC_{50}$ inhibitory concentration of D88 against the production of 2-AA, HHQ, PQS, DHQ, HQNO, and AA by testing gradually increasing concentrations (0.048–50 µM) of this compound. $IC_{50}$ measurements show D88 exerts 50% inhibition against HHQ at a concentration of 3.1 µM, PQS at a concentration of 2.1 µM, HQNO at a concentration of 1.3 µM, 2-AA at a concentration of 1.5 µM, and DHQ at a concentration of 4.14 µM (Fig. 5a, b).

The binding of MvfR protein to the *pqs* promoter is essential for activation of *pqs* operon genes transcription and the subsequent production of 60 small molecules, including the signaling molecules PQS, HHQ, and 2-AA. To determine the efficacy of D88 to disrupt MvfR binding to *pqs* operon promoter and its ability to antagonize with PQS, one of the natural ligands of MvfR[33], we used PA14 cells expressing MvfR fused to a vesicular stomatitis virus glycoprotein (VSV-G) epitope at the C-terminus grown with and without D88 at a concentration of 50 µM. The MvfR–DNA complex was isolated via chromatin immunoprecipitation (ChIP). Quantifying the co-precipitated DNA by qPCR using *mvfR* promoter-specific primers (*pqsA*) shows ~80% reduction of MvfR binding to the *pqs* operon promoter in the presence of 50 µM D88 as compared to the control (Fig. 5c). Moreover, D88 significantly decreased the PQS-mediated increase in MvfR binding when PQS was also added exogenously (Fig. 5c).

Molecular docking analysis using AutoDock Vina[50] reveals that D88, although structurally distinct from MvfR's native ligands PQS and HHQ, targets the same hydrophobic pocket in its ligand-binding domain (LBD) (Fig. 5d) as its ligands and the previously identified BB competitive inhibitor M64[51] (Supplementary Fig. 5). Cross-docking validation indicated that both M64 and D88 are docked into a nearly identical location as M64, shown in the co-crystal structure of the MvfR-M64 complex (Supplementary Figs. 5 and 6). Figure 5D shows a representative structure of the MvfR-D88 docking form with the highest score. The calculated binding energy for MvfR-D88 docking is −10.857 kcal/mol. As shown in Fig. 5e, hydrophobic interactions with Val170, Leu189, Ile236, and Ile263, and pi interaction between Tyr258 and the phenoxy group of D88 appear to be important in MvfR-D88 interaction.

The potential toxicity of D88 was also assessed by utilizing four different cell lines. Cell viability of the human cell lines, hepatoma Hep G2, colorectal adenocarcinoma Caco-2, lung carcinoma epithelial A549 cells, and the mouse macrophage cell line RAW 264.2 was assessed in the presence and absence of D88 at various concentrations. As shown in Supplementary Fig. 7, no significant changes in cell viability were detected after 24 h in any of the cell lines with any of the D88 concentrations tested compared to the vehicle control.

Finally, the drug metabolism and pharmacokinetics (DMPK) of D88 was performed to assess its half-life and bioavailability. Healthy animals received 1 mg/kg intravenous (IV) and 10 mg/kg subcutaneous

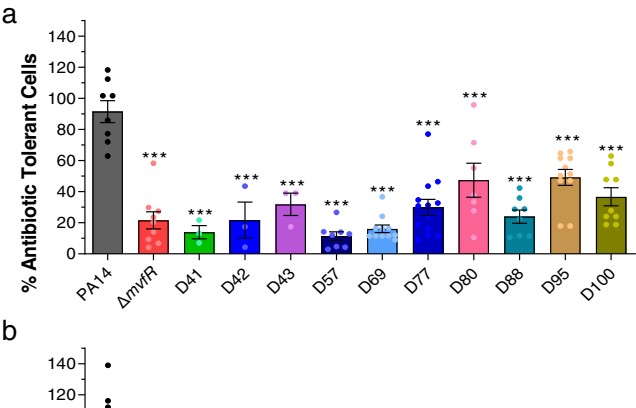

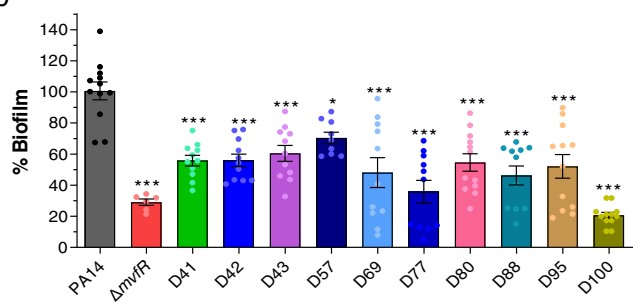

**Fig. 3 | NAMs are potent agents against the formation of antibiotic tolerant, persisters (AT/P) cells and biofilm. a** AT/P cells formation in PA14 and isogenic *ΔmvfR* in the presence or absence of the indicated compounds. Cells were grown in the presence of 10 µg mL⁻¹ meropenem with or without 10 µM of the compounds for 24 h. Values were normalized to the cells grown for 4 h in the absence of antibiotics and compound. The PA14 cells grown with antibiotics and vehicle (DMSO) were considered control, and the percentage values were calculated compared to control. **b** Initiation of biofilm formation of PA14 and *ΔmvfR* cells with or without compound. Biofilm was grown in the 96 well microtiter plate at 37 °C for 24 h containing M63 minimal media in the presence of 10 µM of the compounds or vehicle. After 24 h, the wells were washed to remove planktonic cells, and the biofilm was stained with 0.1% crystal violet. The stained biofilm was washed and solubilized in ethanol: acetone (80:20). OD was measured at 570 nm. The biofilm grown with the vehicle was considered as a control. The percentage value was calculated in comparison to the PA14 control. **a, b** Data represent at least $n = 3$ biological replicates, each dot on the graph represents one replicate, and the number of biological replicates for each compound is depicted in the graph. The error bars denote ±SEM. One-way ANOVA followed by Tukey post-test was applied. ∗, ∗∗ and ∗∗∗ indicate significant differences from the control at $P < 0.05$, $P < 0.01$, and $P < 0.001$, respectively.

(SC) administration of D88, and the half-life and bioavailability of the compound were assessed in the plasma at various time points. Supplementary Fig. 8 shows the D88 half-life ($T_{1/2}$ [h]) (0.27 h and 1.25 h), $T_{max}$ (0.08 h and 0.33 h), $C_{max}$ (1086 ng mL⁻¹ and 1096 ng mL⁻¹), and $C_{last}$ (14.2 ng mL⁻¹, and 66.6 ng mL⁻¹) in plasma following IV and SC administration respectively.

## MvfR promotes intestinal permeability, and its pharmacologic inhibition mitigates the host intestinal barrier damage

Considering the importance of intestinal barrier function in health and the significance of this pathogen as a cause of mortality in critically ill patients, we interrogated the MvfR impact on intestinal permeability and the efficacy of the D88 compound in interfering with MvfR function. We used paradigmatically a well-established clinically relevant infection mouse model (Fig. 6a) as a surrogate of the critical illness status and gut permeability, allowing us to test the systemic consequences of *P. aeruginosa* infection initiating at a distant site such as a wound site. Given that the extensive Total Burn Surface Area (TBSA) and the high inoculum used in this model exemplify the infection-related adverse outcomes in the host following acute *P. aeruginosa* infections in critically ill patients. Therefore, the results of these in vivo experiments could be relevant in the setting of *P. aeruginosa* infections

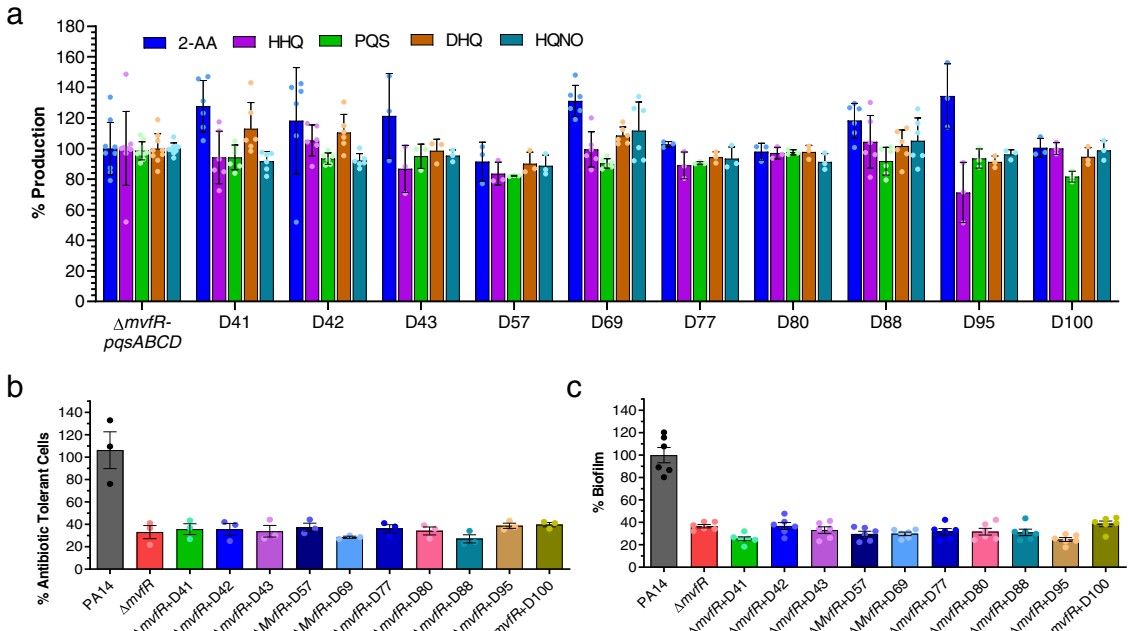

**Fig. 4 | In vitro assessment of NAMs efficacy indicates no off-target effect.**
**a** Effect of the selected compounds on the MvfR-independent HAQs, 2-AA, and DHQ production. The production of the MvfR-regulated small molecules 2-AA, HHQ, PQS, DHQ, and HQNO was measured in the cultures at $OD_{600nm} = 3.0$ in the presence of 50 μM of the indicated compounds using a *MvfR* mutant strain that constitutively expresses the *pqsABCDE* genes. The cells were grown with or without (vehicle only) compound, and the small molecules production was measured using liquid chromatography-mass spectrophotometry (LC-MS). The percentage production was calculated compared to cells grown with the vehicle control. **b** AT/P cell formation in the PA14 isogenic mutant *ΔmvfR* in the presence of the indicated compounds. Cells were grown in the presence of 10 μg mL$^{-1}$ meropenem, with or without 10 μM of the compounds for 24 h. Values were normalized to cells grown for 4 h in the absence of antibiotics and compounds. The *ΔmvfR* cells grown with

antibiotics and vehicles were considered control, and the percentage values were calculated compared to control. **c** Biofilm formation of mutant *ΔmvfR* cells with or without compound. Biofilm was grown in the 96-well microtiter plate at 37 °C for 24 h containing M63 minimal media in the presence of 10 μM of the compounds or vehicle. The biofilm grown with the vehicle was considered as a control. The percentage value was calculated in comparison to the PA14 control. **a–c** Data represent at least $n = 3$, each dot on the graph represents one biological replicate. The number of biological replicates for each compound and strain is depicted in the graph. The error bars denote ±SEM. HAQs = 4-hydroxy-2-alkylquinolines; 2-AA = 2-aminoacetophenone; DHQ = 2,4-dihydroxyquinoline; PQS = 3,4-dihydroxy-2-heptylquinoline; DHQ = 2,4-dihydroxyquinoline; HQNO = 4-hydroxy-2-heptylquinoline *N*-oxide.

---

that are related to any type of critical illnesses, as delineated in the introduction.

Although MvfR function is required for full virulence in vivo, its direct role in intestinal permeability has not been demonstrated directly. We measured the FITC-dextran 3–5 kDa flux from the intestinal lumen to the systemic circulation at 22 h following burn and infection using the specific regiment shown in Fig. 6a. Flux differences between the groups were determined at 22 h when the burn impact on gut permeability essentially returns to the sham levels, while the strong effect of infection on the intestinal barrier dysfunction is still observed. Figure 6b shows that the mice that were burned and infected with PA14 exerted a higher flux of FITC-dextran out of the intestinal lumen (mean FITC-dextran = 4639 ng mL$^{-1}$) compared to the mice that only underwent burn injury without infection (mean FITC-dextran = 1158 ng mL$^{-1}$; $P < 0.001$). In contrast, mice infected with the PA14 isogenic *mvfR* mutant displayed a significantly decreased intestinal barrier dysfunction (mean FITC-dextran = 1179 ng mL$^{-1}$), clearly showing the role of MvfR in intestinal permeability.

The efficacy of the highly favorable profile of the D88 compound in vitro is reproduced in vivo. Figure 6b shows that D88 significantly ameliorated the intestinal barrier dysfunction in mice following burn and infection, reducing the FITC-dextran flux in the burnt and infected mice (mean FITC-dextran = 2759 ng mL$^{-1}$; $P < 0.01$), as compared to the vehicle-treated group, which exerted a similar phenotype as the animals infected with wild type PA14 (mean FITC-dextran = 4611 ng mL$^{-1}$). These findings demonstrate that the D88 effect in the treatment group in mitigating the PA14-mediated derangement of intestinal permeability is solely attributed to the compound itself.

Moreover, to determine whether D88 has any potential off-target effect in vivo, we administered D88 to mice infected with the PA14 isogenic *mvfR* mutant strain (Fig. 6b). As expected, there was no difference in the intestinal permeability status between the *mvfR* mutant-infected mice treated with D88 and the *mvfR* mutant infected ones that received no treatment or those that received the vehicle control (Fig. 6b). Controls were represented by two additional groups, where mice only underwent burn injury without any subsequent infection and administration of D88 or the vehicle control. None of these treatments altered the phenotype that we observed in the burn-alone group of mice, with the level of FITC-dextran detected in the systemic circulation being the same as in the burnt mice that did not receive any treatment (mean FITC-dextran = 1158 ng mL$^{-1}$ for the no treatment group; mean FITC-dextran = 850 ng mL$^{-1}$ for the D88-treated group) (Fig. 6b).

### D88 ameliorates bacterial dissemination to the small and large intestine

Intestinal hyperpermeability in bacterial infections that originate outside the intestinal lumen has previously been correlated with an increased systemic bacterial load. Therefore, given the aforementioned intestinal alterations in terms of function and morphology following infection, we determined the bacterial dissemination from the site of infection to the small and large intestine of the mice in this setting. As shown in Fig. 6c, PA14 disseminated in distant organs (ileum and colon) in higher numbers than the isogenic *mvfR* mutant strain, indicating the significant role of the MvfR function in vivo in the ability of bacteria to disseminate in different host organs. When D88 was administered, bacterial dissemination to the intestinal tissues was also

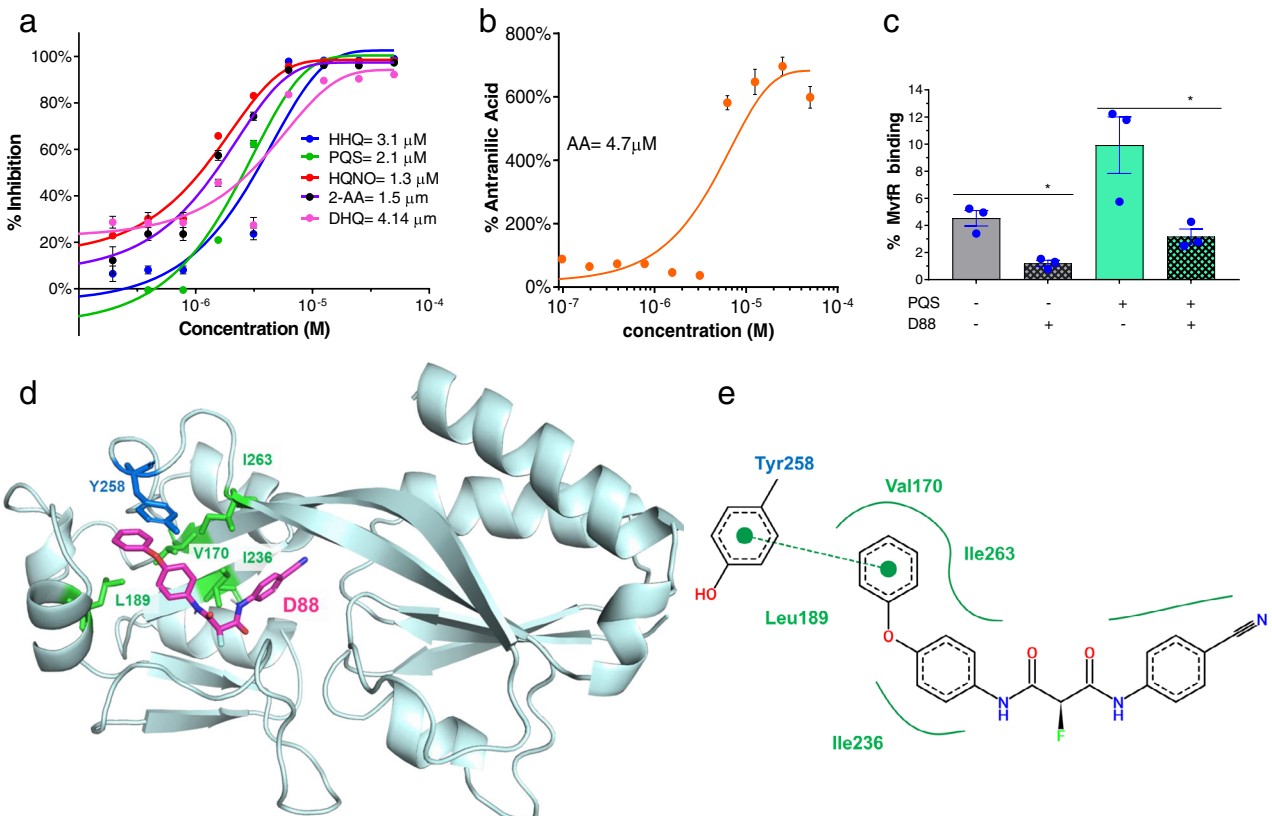

**Fig. 5 | IC$_{50}$ measurements, in vitro engagement, and molecular docking studies support the prioritization of compound D88. a** Dose-dependent inhibition of HHQ, PQS, HQNO, 2-AA, and DHQ production was measured in PA14 cultures at OD$_{600nm}$ = 3.0. The cells grown with the compound's vehicle were considered the control. **b** Dose-dependent production of AA in the presence of D88 in PA14 cultures at OD$_{600nm}$ = 3.0. AA is the primary precursor of all the MvfR-regulated small molecules assessed in A. GraphPad Prism 9.2.0 software plotted the IC$_{50}$ curves against percent inhibition of the HAQs, 2-AA, DHQ, and AA production at each concentration. **a, b** Data represent the mean value of $n = 3$ biological replicates. Error bars denote ±SEM. **c** D88 inhibition of MvfR binding to the *pqsA* promoter. Overnight grown culture of PA14 expressing MvfR-VSV-G was diluted to OD 600$_{nm}$ 0.01 and grown at 37 °C with and without D88 (50 μM) and/or PQS (38 μM) until OD 600 $_{nm}$ 1.0. Thereafter MvfR-DNA complex were cross-linked and isolated via chromatin immunoprecipitation (ChIP). Coprecipitated DNA was purified and quantified using quantitative real-time polymerase chain reaction qPCR. MvfR binding to the *pqsA* promoter was calculated using the input method. Data reprsent $n = 3$ biological replicates, and the error bars denote ±SEM. Statistical analysis was carried out using GraphPad Prism 9.2.0 software. Unpaired $t$ test was applied between the compared group. * indicate significant differences from the control at $P < 0.05$ (Control vs. PQS = 0.038; PQS vs. PQS + D88 = 0.012). **d** The representative structure of the MvfR ligand-binding domain (LBD) complex with D88 (magenta sticks) was obtained from the docking analysis using AutoDock Vina. Tyr258 related to pi-interaction and Val170, Leu189, Ile236, and Ile263 related to hydrophobic interaction are shown as blue and green sticks, respectively. **e** Two-dimensional diagram of MvfR-D88 docking analyzed using PoseView program. A Green dashed line connecting two green dots indicates pi interaction. The solid green line indicates hydrophobic interactions made by hydrophobic residues (Val170, Leu189, Ile236, and Ile263) surrounding D88. AA = anthranilic acid; HHQ = 4-hydroxy-2-heptylquinoline; PQS = 3,4-dihydroxy-2-heptylquinoline; HQNO = ; 2-AA = 2-aminoacetophenone; DHQ = 2,4-dihydroxyquinoline; MvfR-VSV-G = MvfR fused to a vesicular stomatitis virus glycoprotein; Val = ; LeTyr = Tyrosine; Val = Valine; Leu = Leucine; Ile= Isoleucine.

significantly reduced. Administration of the vehicle control did not affect bacterial dissemination in the ileum or colon (Fig. 6c).

In the same set of experiments, we assessed the bacterial load at the site of infection. Figure 6c shows that the bacterial load at the inoculation site was the same between the PA14 and the *mvfR* mutant, and it was not affected by either treatment (D88 or the vehicle control). These results corroborate our findings regarding the MvfR role in bacterial virulence in vivo rather than viability and indicate that inhibition of MvfR confers significant protection from systemic bacterial dissemination.

**D88 shows strong target engagement in vivo**

To determine the ability to engage with the target in vivo, we assessed the levels of HHQ, PQS, HQNO, 2-AA, and DHQ in D88 treated and untreated mice at the site of infection where the bacterial burden was similarly high as in PA14 and *mvfR* (Fig. 6c). Figure 6d shows the strong inhibitory efficacy of D88 at 12 h with a complete abolishment of these MvfR-regulated virulence-related molecules at 22 h post-infection even though the bacterial load at the site of infection was as high as

$1 \times 10^8$ in the infected and D88 treated animals and almost identical to the CFUs of the infected + vehicle (untreated) mice (Fig. 6c). This finding demonstrates the anti-virulence efficacy of D88 and indicates its strong engagement to MvfR in vivo over time.

**D88 mitigates the morphologic alterations of the intestinal lining and attenuates intestinal inflammation**

Key regulators of the intestinal barrier function are multi-protein Tight Junction (TJ) complexes that orchestrate the paracellular intestinal permeability. Claudins and zonula occludence junctional proteins are critical modulators of intestinal barrier integrity[52]. More specifically, some junctional proteins are protective 'tightening' proteins, while others mainly contribute to intestinal permeability functions[53]. We sought to evaluate the changes of one such TJ protein, claudin-1, by exposure-matched confocal microscopy images and subsequent intensity quantification analysis. For these studies, we used ileum samples from mice that underwent burn and infection with PA14 or with the isogenic *mvfR* mutant strain in the presence or absence of D88. Figure 7a and Supplementary Fig. 9 show a marked decrease in

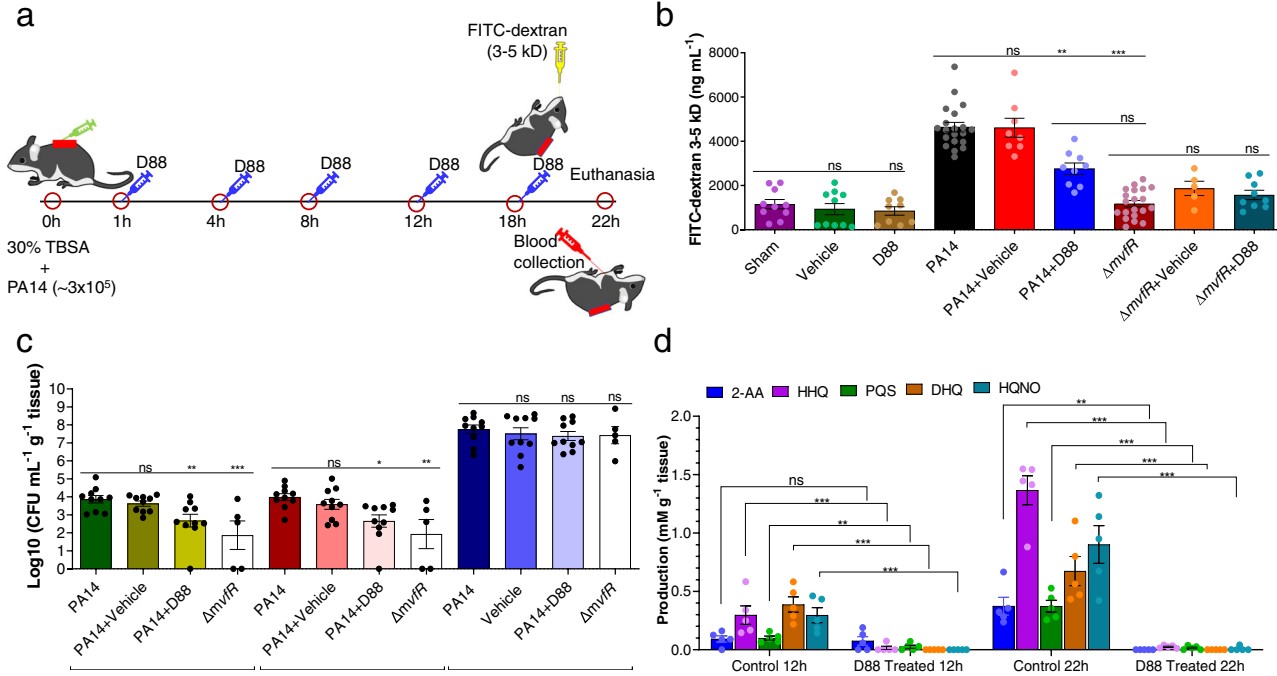

**Fig. 6 | MvfR promotes intestinal permeability. Its pharmacologic inhibition mitigates the host intestinal barrier damage, ameliorates bacterial dissemination, and abolishes the production of the small molecules. a** Schematic representation of the burn-site infection model and treatment plan. The color of syringes indicates administration: Green bacterial inoculum; blue D88; yellow FITC-dextran; and red blood collection. **b** Fluorescein Isothiocyanate-Dextran (FITC-dextran) 3-5 kDa levels in the serum 22 h post-infection. FITC-dextran 3-5 kDa was gavaged 18 hr following burn and infection. Blood was collected 4 h following gavage (22 h post-burn and infection), and the FITC-dextran fluorescence intensity was measured using fluorescent spectrophotometry (excitation, 480 nm, and emission, 520 nm). **c** Effect of D88 on the bacterial dissemination to the ileum and colon and bacterial load at the site of infection. Small and large intestinal tissues and muscle underlying the burn eschar and infection site from mice of each group were collected at 22 h post-burn and infection. Sample homogenates were serially diluted and plated on Pseudomonas isolation agar plates. Bacterial colony-forming units (CFUs) were counted and normalized by the tissue weight. **d** D88 inhibits the production of PQS, HHQ, HQNO, DHQ, and 2-AA in the infected mice. Production was measured in samples of the underlying muscle at the site of infection. Tissue was collected at 12 h and 22 h post-infection, and these molecules were quantified using liquid chromatography-mass spectrophotometry (LC-MS). **b–d** Data represent at least $n = 5$ biological replicates, each dot represent data from one mouse. The exact number of mice used in each condition is indicated in each bar. The error bars denote ±SEM. **b–d** One-way ANOVA followed by Tukey post-test was applied. The no-treatment group data (Burn + PA14) were compared to the vehicle and the D88 treated groups. *, **, and *** indicates significant differences compared to the control at $P < 0.05$, $P < 0.01$, and $P < 0.001$, respectively. ns represents no significant difference.

staining for claudin-1 in the PA14 + vehicle infection group that received no treatment (60.41 fluorescence intensity arbitrary units (AUs)), as compared to the mice that were infected with PA14 and received D88 (142.48 AUs), as well as compared to animals that were infected with the isogenic *mvfR* mutant strain (144.58 AUs). The images show a less even distribution at the areas of cell-cell contact and an eliminated delineation of the cell periphery in the setting of PA14 infection in the absence of MvfR inhibition. On the contrary, both the *mvfR* mutant and treatment with D88 attenuate these effects, as can be appreciated in the results shown in Fig. 7a. The staining for claudin-1 exhibits a more organized appearance at the periphery of the cells, with a more uniform localization at the sites of cell-cell interaction. These data indicate a considerable improvement in the morphology of the intestinal paracellular transport following MvfR silencing.

Following derangement of the intestinal barrier integrity, microbial paracellular transport out of the lumen cues an inflammatory response from the intestinal mucosa[54]. Similarly, mucosal inflammation is known to increase the TJ disruption-mediated permeability, further deranging the paracellular transport[55], leading to a vicious cycle of defective intestinal integrity. Therefore, we determined whether and how MvfR inhibition attenuates the changes in the levels of intestinal inflammation in our mouse model. Figure 7b demonstrates a sharp rise of the ileal tumor necrosis factor (TNF-α) in the group that was infected with PA14 and received no treatment (mean TNF-α level of 519 pg mL⁻¹ g⁻¹ tissue) or received the vehicle control (mean TNF-α

level of 504 pg mL⁻¹ g⁻¹ tissue), as compared to the burn alone group ($P < 0.0001$). D88 treatment confers a significant reduction in the TNF-α levels (mean of 381 pg mL⁻¹ g⁻¹ tissue; $P < 0.05$). The group infected with the isogenic *mvfR* mutant strain exhibited an even lower level of TNF-a, with the mean concentration being 149 pg mL⁻¹ g⁻¹ tissue ($P < 0.0001$).

Similarly, we investigated the changes in the levels of ileal interleukin-6 (IL-6), which displayed a significant rise in the ileum of PA14-infected mice ($P < 0.001$) and a marked decrease in the D88 administration group and the *mvfR* mutant infection groups ($P < 0.05$ and $P < 0.001$, respectively) (Fig. 7c). No differences are observed between the *mvfR* infected treated and untreated animals, further demonstrating that mitigation of inflammation is not attributed to an off-target effect. The observed differences in the TNF-a and IL-6 levels further support our observation that MvfR inhibition in vivo significantly diminishes inflammation within the intestinal lumen. These data together highlight the importance of MvfR silencing in vivo as a therapeutic strategy in the setting of *PA* infections.

## Discussion

*P. aeruginosa* colonizes the intestinal tract and aggravates the derangement of the intestinal barrier in critically ill patients who have defective intestinal integrity secondary to their primary clinical condition[12,13]. Here we demonstrate that MvfR (PqsR) represents an excellent target for limiting the ability of this pathogen to promote

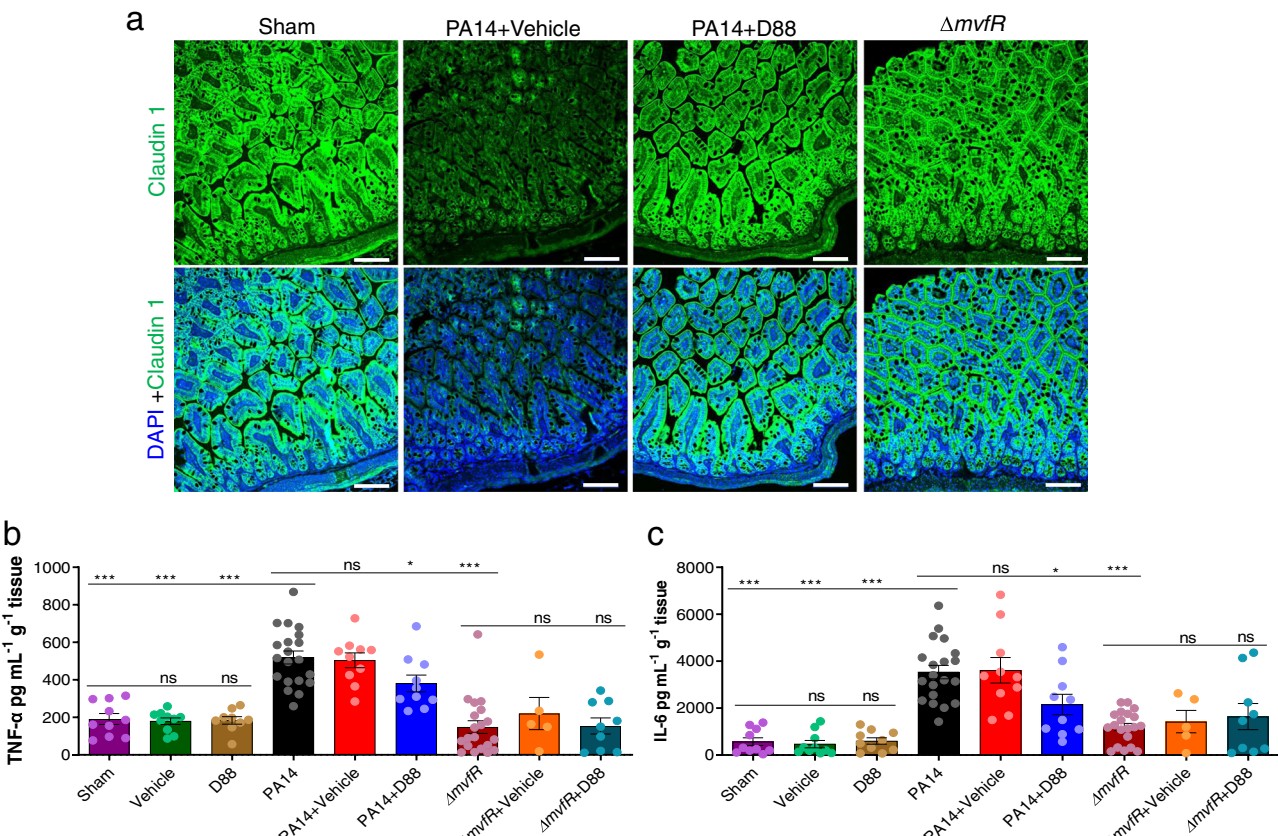

**Fig. 7 | D88 mitigates the morphologic alterations of the intestinal lining and attenuates intestinal inflammation. a** Representative confocal image ($n = 2$) of distal ileum with Claudin-1 immunofluorescence staining. The experiment was repeated independently two times with similar results. Samples for confocal imaging were harvested 22 h post-burn and infection. Green fluorescence represents Claudin-1, and blue fluorescence represents the DAPI stain, the white line represents the scale bar (100 μm). **b** Levels of tumor necrosis factor (TNF-α) in the distal ileum 22 h post-burn and infection. The total protein was isolated from the distal ileum, and the concentration of TNF-α in the sample was quantified using ELISA. **c** The levels of interleukin (IL-6) in the distal ileum were also quantified using ELISA. **b**, **c** Data represent at least $n = 5$, each dot in the bars represents data from one mouse. The error bars denote ± SEM. One-way ANOVA followed by Tukey post-test was applied. The no-treatment group data (Burn + PA14) were compared to the vehicle-treated and the D88-treated groups. ∗, ∗∗, and ∗∗∗ indicates significant differences compared to the control at $P < 0.05$, $P < 0.01$, and $P < 0.001$, respectively. ns represents no significant difference.

intestinal barrier derangement. This work is the first to report the activity of a NAM family of potent and effective anti-MvfR agents with a structure unrelated to the MvfR ligands/inducers PQS and HHQ and has no apparent chemical liabilities for in vivo use. The extensive SAR studies performed clearly show that the presence of an electron-donating phenoxy group on one of the aromatic rings leads to very active compounds. In contrast, adding a substituent to the methylene group generally results in inactive compounds except with fluorine which generates very active compounds such as the highly soluble D88 (Supplementary Tables 1–8 and Fig. 2). Overall, many NAM derivatives show much better solubility than the benzothiazole analogs we previously reported[22].

Several groups studying *P. aeruginosa* MvfR inhibition have so far reported compounds that show MvfR-inhibitory effects, as indicated by the repression of MvfR-regulated genes and functions in vitro. Among them, several plant and bacterial extracts and synthetic compounds have been used as anti-MvfR agents[56–58]. Recently Grossman et al. reported on the activity of the thiazole-containing quinazolinones inhibiting MvfR function, but this was limited to the inhibition of pyocyanin in vitro[59]. The inhibitor, 2-((5-methyl-5H-[1,2,4]triazino[5,6-b]indol-3-yl) thio) showed efficient inhibition, and the IC$_{50}$ value was low; however, this compound was only tested for the inhibition of pyocyanin, HHQ, and PQS production in vitro. Another MvfR antagonist, a thioether-linked dihydropyrrol-2-one analog, was reported to inhibit biofilm formation[58]. Generating non-ligand-based MvfR-

inhibiting compounds is of great importance since *P. aeruginosa* can modify ligand-based MvfR inhibitors into *MvfR* activators[60], which could ultimately increase virulence.

Especially relevant in our study is the in vivo monotherapy efficacy of our anti-MvfR compounds in a vertebrate *P. aeruginosa* infection model. Despite the extensive studies in *P. aeruginosa* QS inhibition over the years, there are few studies that report in vivo effects with anti-MvfR agents. The available in vivo studies that assess the impact of anti-MvfR compounds have been limited to testing these agents in non-vertebrate model hosts[61]. Though these are significant findings, the results in these model organisms might not be directly translatable to humans. Determining whether QS inhibitors are indeed effective in vivo and monotherapy is crucial in determining if such compounds have the potential to be incorporated in preventive or treatment approaches and strategies against *PA* infections in humans in the future. Most recently, Schütz et al. reported the inverse agonist QSI 4 for their anti-MvfR activity; the compound showed efficacy in vitro and has suitable pharmacokinetics (in a murine model); interestingly, however, the compound has antibacterial activity[62]. The 4-log reduction of the bacterial load at the murine infection site is the opposite of the characteristic of anti-MvfR compounds, and the lack of assessment of potential off-target effect might imply that an additional QSI target might be responsible for this reduction.

Our findings also indicate that D88 is highly efficient in target engagement in vitro and in vivo. The compound appears not to have an

off-target effect in the murine model tested, as shown by its efficacy in the setting of infection with the isogenic *mvfR* mutant in mice in the presence of D88, and does not target the activity of any of the *pqsABCD* operon encoded enzymes or changes MvfR-regulated phenotypes in the setting of the *mvfR* mutant carrying the constitutively expressed *pqsABCD* operon.

The in vivo results show that D88 monotherapy confers significant protection against the expected *P. aeruginosa*-mediated intestinal permeability derangement. This is a significant finding given that the animal model used here represents an acute infection model in the setting of critical illness (severe burn in this instance). These results are also well correlated with the lower levels of inflammatory cytokines in the small intestine of the mice following D88 treatment compared to the no-treatment group. In addition, despite the same bacterial load at the site of infection in all groups, we observed reduced bacterial dissemination to the intestinal tissues when MvfR was inhibited and abolishment of the MvfR-regulated small molecules, including the MvfR inducers and signaling molecules, underscoring the impact of its pharmacologic inhibition and the importance of treating site wound infections. Reduced inflammation levels and diminished bacterial dissemination in the intestinal tissue can justify well the improved intestinal barrier function observed following compound administration.

Notably, this study carries two additional features related to the potential use of the new class of MvfR inhibitors as anti-*Pseudomonas* therapeutics. First, the herein-reported compounds are highly effective against multiple clinical *PA* isolates, showing the breadth of their efficacy beyond the PA14 strain. Second, the compound D88 reported here bears no substituents incompatible with in vivo use and exhibits no toxic effect following their use in macrophage and liver hepatoma cells and lung and intestinal epithelial cells. The transfer of QS inhibitory compounds to clinical practice has so far been significantly limited by their cytotoxic effects or their unfavorable pharmacological properties[63]. Therefore, the favorable profile of our compounds for in vivo use underscores the impact and significance of this work.

Further research will be needed to address additional questions. For instance, whether the reported compounds are efficient in other infection model settings and could be used as effective adjuvants to antibiotic treatments against *PA* infections. Combining anti-MvfR agents with antibiotics may aid in reducing antibiotic doses, which could subsequently reduce the selective pressure on the pathogen to develop resistance. Nevertheless, just the loss of MvfR function completely abolishes many acute and chronic virulence-related functions[25,34–36,38,40,44–47], such as those promoted by 2-AA, i.e., *lasR* mutations[32] and AT/P cells formation[31], which also dysregulates the host immune mechanisms[28,39,47] and host chromatin modifications, leading to persistent infections[28,31,32,64].

Furthermore, even though our in vivo results were obtained with a vertebrate infection model, it would be useful to determine the effect of our compounds in a setting that more closely resembles human homeostasis, pathology, and pathophysiology. Recent advances in the field of organ-on-chip technology emulating native tissue architecture and the mechanochemical environment of the human tissues could be of help. Nevertheless, this work highlights the importance of maintaining gut mucosal integrity as part of any successful strategy to prevent/treat infections and the gut-derived sepsis syndrome seen in critically ill patients. Our study opens new avenues for the care of these patients.

## Methods

### Synthesis and structure modification of NAMs

Synthesis of the compound D88 was performed in three steps (Supplementary Fig. 10): i). to a stirred solution of compound 1 (9 g, 0.059 mol) in dichloromethane (250 mL), compound 2 (11.1 g, 0.059 mol), EDC (13.8 g, 0.071 mol), N, N-diisopropylethylamine (17 g, 0.13 mol) were added. The resulting solution was stirred at room temperature for 16 h. Then the reaction mixture was evaporated to dryness. The product was purified by column chromatography to give compound 3 (5 g, 26% yield). ii). To a stirred solution of compound 3 (5 g, 0.0157 mol) in THF and water (1:1) (200 mL), lithium hydroxide monohydrate (1.3 g, 0.031 mol) was added in one portion. The resulting mixture was stirred at room temperature for 12 h. Then the reaction mixture was acidified with 3 N HCl, and the solvent was removed in vacuo; the solid was filtered, washed with $H_2O$, and dried in vacuo to give compound 4 (4.5 g, 98% yield). iii). To a stirred solution of compound 4 (4.5 g, 0.015 mol) in DMF (150 mL), compound 5 (1.77 g, 0.015 mol), HATU (6.84 g, 0.018 mol), N, N-diisopropylethylamine (2.3 g, 0.018 mol) were added. The resulting solution was stirred at room temperature for 16 h. Then the solvent was evaporated to dryness. The product was purified by column chromatography to give the title compound D88 (1.7 g, 30% yield). The route of synthesis for compound D88 is shown in Supplementary Fig. 10. Analytical data showing 1H NMR and Mass spectrometry spectra of non-commercially available compounds are shown in Supplementary Fig. 11a–h. Certificate of analysis reports compounds' purity ~ 95%. Compounds' syntheses, including D88 was performed by Enamine Ltd., Kyiv, Ukraine.

### Bacterial strains and growth conditions

UCBPP-PA14 (PA14) is a rifampicin-resistant *P. aeruginosa* human clinical isolate (Rahme laboratory and Rahme et al., 1995)[65]. The *mvfR* mutant is isogenic to UCBPP-PA14 (Rahme laboratory)[65]. The ΔmvfR-pPqsABCD strain, which constitutively expresses the *pqs* operon expression, was generated by cloning the *pqsABCD* operon into pDN18 and electroporating this construct into PA14 Δ*mvfR* cells (Rahme laboratory and Starkey et al., 2014)[22] P*pqsA*-GFP_{ASV} was previously described[48]. The *P. aeruginosa* clinical isolates LGR-4325, LGR-4326, LGR-4327, LGR-4328, LGR-4330, LGR-4331, LGR-4333, LGR-4334, LGR-4343, LGR-4344, LGR-4348, LGR-4356, LGR-4362, LGR-4363, LGR-4364, and LGR-4366 were obtained from Shriners Hospital for Children Boston, Boston MA. Unless otherwise indicated, bacteria were grown in Lysogeny Broth (LB) broth, LB agar plates, or LB agar plates containing 100 μg mL$^{-1}$ rifampicin.

To start each assay, unless otherwise specified, bacterial cells were streaked from a −80 °C stock on an LB agar plate at 37 °C. A single bacterial colony was then inoculated in LB medium and grown at 37 °C overnight and used as a starter culture diluted 1:1000 for an over-day culture grown at 37 °C to the desired optical density (OD) for the assay used.

### Pyocyanin production assay

The over-day culture was grown to $OD_{600nm}$ 1.5., diluted again at 1:10,000 in 5 mL LB and incubated with 10 μM of the test compound or the vehicle control for 18 hours at 37 °C and 200 rpm in an incubator shaker. After 18 h, 1 mL of culture was centrifuged for 2 min at 20.000 x *g*. The supernatant was transferred to a new tube, 200 μL of this were loaded in a flat-bottom 96-well plate, and the pyocyanin levels were quantified by measuring at $OD_{690nm}$ in the infinite F200 plate reader using the Magellan software (Tecan, Switzerland). The same procedure was also followed for the pyocyanin production assessment for all the *P. aeruginosa* clinical isolates assessed in this study. The percent of pyocyanin was calculated compared to the PA14 culture grown in the absence of compounds.

To determine the $IC_{50}$ of pyocyanin production, we followed the same process as above. Dose-dependent inhibition of pyocyanin production was performed by incubating the bacterial cultures with 13 different concentrations of each tested compound (50 μM, 25 μM, 12.5 μM, 6.25 μM, 3.125 μM, 1.562 μM, 0.781 μM, 0.390 μM, 0.195 μM, 0.0976 μM, 0.0488 μM, 0.0244 Mm and 0.0122 μM). Using GraphPad Prism Software, the $IC_{50}$ curve was plotted against percent inhibition

of pyocyanin production at each concentration using PA14 in the absence of compounds as a control.

## Effect of compounds on the expression of *pqsA-GFP*

The expression levels of *pqsA*, which is representative of the *pqsABCDE* operon transcription, were monitored in the presence of the MvfR inhibitors using PA14 cells containing the P*pqsA*-GFP$_{ASV}$, a reporter construct of the pqsA promoter fused to a short half-life GFP gene in, such that quantitative quenching of fluorescence corresponds to *pqsA* promoter repression[22,32,48]. The PA14 P*pqsA*-GFP$_{ASV}$ cells were grown in 96 well plates in the presence of 10 μM of the inhibitory compounds and incubated at 37 °C in the Infinite F200 plate reader using the Magellan software (Tecan, Switzerland). GFP fluorescence levels were measured at $\lambda_{ex} = 485\,_{nm}/\lambda_{em} = 535\,_{nm}$ after 10 seconds of shaking every 15 min up to 12 h. The IC$_{50}$ of P*pqsA-GFP* expression levels was performed as above using the same 13 different concentrations (0.0122–50 μM) for each of the 13 compounds tested. The IC$_{50}$ curve was plotted against percent inhibition of the *pqsA-GFP* expression at each concentration using GraphPad Prism 9.2.0 software.

## Antibiotic Tolerant/ Persister (AT/P) cells formation assay

The AT/P cell formation assay was performed according to the method previously described in Starkey et al., 2014 (persister cell assay). The over-day culture was incubated at 37 °C with shaking at 200 rpm until the culture OD$_{600nm}$ was 3.0. The culture was then diluted at 1:100 and grown in the same conditions for 4 h. After 4 h, 200 μL were collected, and serial dilutions were plated on LB agar plates for CFU quantification (as normalizers). The remainder of the culture was treated with meropenem to a final concentration of 100 × MIC (Minimum Inhibitory Concentration; 10 mg L$^{-1}$) and compounds at a final concentration of 10 μM. After 24 h, culture aliquots were 10-fold serially diluted in LB broth and plated on LB agar for CFU quantification. The AT/P fraction was determined as the normalizers (pre-antibiotic) ratio divided by the AT/P fraction (24 h post-antibiotic)[22].

## Assessment of biofilm formation in 96-well plates

The overnight grown starter culture was diluted 1:1000 in 5 mL LB media in a glass tube and was incubated at 37 °C and 200 rpm in an incubator shaker until the culture reached an OD$_{600nm}$ 3.0. The culture was then diluted at 1:100 in M63 minimal media supplemented with 0.2% glucose, 0.5% Casamino Acids, and 1 mM MgSO$_4$. Initiation of the Biofilm formation was performed in the 96 well plates; 200 μL of diluted culture was added in the wells with (10 μM) or without compounds. The cultures were allowed to grow at 37 °C in static conditions for 24 hours. Thereafter, planktonic culture was removed from the wells, and wells were washed three times with distilled water (DW). Biofilm was stained with 0.1% crystal violet and incubated for 15 min at room temperature (RT). Access dye was removed from the wells and washed off three times with DW, and 200 μL ethanol: acetone (80:20) was added to the wells. The plate was incubated for 30 min at RT, and the OD was measured by spectrophotometry at 570 nm in the infinite F200 plate reader using the Magellan software (Tecan, Switzerland). Percent biofilm formation was calculated in comparison to PA14 culture that was grown in the absence of compounds.

## Production of 4-hydroxy-2-alkylquinolines (HAQs), 2-aminoacetophenone (2-AA), 2,4-dihydroxyquinoline (DHQ)

The HAQs HHQ, PQS, HQNO, and non-HAQs: 2-AA and DHQ produced by PA14 cells were quantified in the presence of NAMs. The overnight grown starter culture was diluted 1:100 in 5 mL LB media in a glass tube and was allowed to grow in the presence (50 μM) or absence of compound at 37 °C and 200 rpm in an incubator shaker until OD$_{600}$ was 3. Bacterial cultures were subsequently mixed 1:1 (400 μL: 400 μL) with 100% methanol containing 20 ppm (20 μg mL$^{-1}$) of tetradeutero-PQS (PQS-D4) and 10 ppm of tetradeutero-HHQ (HHQ-D4) in a 1.5 mL

Eppendorf tube. The mixture was vortexed for 5 seconds and spun down for 5 minutes at 12,000 x *g*. A 700 μL aliquot of the supernatant was removed and stored in glass vials at −20 °C until further liquid chromatography-mass spectrophotometry (LC/MS) analysis as previously described in Lepine et al., 2003[66]. LC/MC analysis was performed using a Micromass Quattro II triple quadrupole mass spectrometer (Micromass Canada, Pointe-Claire, Canada) in positive electrospray ionization mode, interfaced to an HP1100 HPLC equipped with a 4.5 × 150 mm reverse-phase C8 column.

## The binding affinity of compounds to the MvfR protein

The selected compounds were tested for their ability to bind to MvfR. Target validation was carried out by using surface plasmon resonance (SPR). The MvfRC87 protein purified as in Xiao et. al., 2006[33] was covalently immobilized on a CM7 Series S sensor chip using an amine coupling reagent kit (GE Healthcare) at the range level of 3000 to 5000 response units (RU). RU was analyzed by Biacore T200 evaluation software 2.0 (GE Healthcare). The relative response units (RUs) 5 seconds before the end of the association were extracted from the double-reference-corrected sensor grams at different concentrations. These responses were plotted against their respective concentrations for MvfR inhibitor alone and in the presence of native MvfR ligand PQS and compared to the calculated responses for the mixture expected for different binding sites[51].

## Measurement of compounds' solubility

Solubility of the 10 mM compounds dissolved in DMSO was tested in the isotonic phosphate buffer (iPBS) at pH 7.4 using High-Performance Liquid Chromatography (HPLC). For the preparation of iPBS phosphate-buffered saline tab (Sigma-Aldrich 08057-12Tab-F) is dissolved in 500 mL of deionized water and the final composition of the solution contains 10 mM PBS, 2.7 mM KCl, 140 mM NaCl and 0.05% Tween. A total 10 μL of the compound from a 10 mM stock solution in DMSO was added to the vial, and 190 μL of buffer was added to each vial and mixed with shaking at room temperature for 4 h. Thereafter the samples were filtered, and 160 μL of the sample was mixed with 40 μL of DMSO for the final injection. Ibuprofen (high solubility) and Progesterone (low Solubility) were used as standard compounds. Ammonium Acetate aqueous solution (50 mM, pH 7.4) and the acetonitrile were used as mobile phases A and B, respectively. The measurement of concentration was achieved by comparing of UV absorbance of the sample solution and the known standard solution following an HPLC separation using a generic fast gradient method. The solubility of each compound is expressed by the ratio of compound amount in the sample test solution to the amount of compound in the standard solution.

## Cytotoxicity assessment

For the determination of cytotoxicity, cell viability was determined using the MTT (3-[4, 5-dimethyl-2-thiazolyl]−2, 5-diphenyl-2H-tetrazolium bromide) assay in the presence and absence of compound (D88). Four different cell types obtained from ATCC, namely RAW 264.7 (macrophage; Catalog#TIB-71), Caco-2 (colon epithelial cells; Catalog# HTB-37), Hep G2 (liver cells; Catalog# HB-8065), and A549 (lung epithelial cells; Catalog# CCL-185), were grown at 37 °C, 5% CO$_2$, in Dulbecco's Modified Eagle Medium (DMEM) or Eagle's Minimal Essential Medium (EMEM) containing 10% Fetal Bovine Serum (FBS), 2 mM glutamine, and antibiotic-antimycotic until 80-90% confluency was reached in 96-well plates. The cells were then treated with either vehicle or different concentrations of D88 (10, 20, 30, 40, and 50 μM) for 24 h at 37 °C and 5% CO$_2$. After 24 h, the cells were washed three times with PBS and were incubated in 200 μL PBS containing 200 μg mL$^{-1}$ MTT (Sigma-Aldrich) in a 96-well culture plate for 2 h at 37 °C and 5% CO$_2$. Following incubation, the supernatant was discarded, and the cells were lysed in 95% isopropanol and 5% formic acid.

Absorbance was measured at $OD_{570nm}$ by spectrophotometry in the infinite F200 plate reader using iControl software (Tecan, Switzerland).

## In Silico docking of MvfR-D88

Molecular docking was carried out using AutoDock Vina integrated with the UCSF Chimera docking program (https://vina.scripps.edu)[50,67]. The structure of the MvfR ligand-binding domain in complex with M64 (PDB ID: 6B8A; https://doi.org/10.2210/pdb6b8a/pdb) was used for docking analyses to examine whether D88 fits into the same pocket where the native ligands and M64 bind[51]. M64 was also used for cross-docking validation. The docking results were visualized on the Pymol software, and the interaction forms of MvfR-D88 were analyzed using the PoseView program (https://proteins.plus/)[68].

## Determination of IC$_{50}$ of the compound D88 for the HAQs, 2-AA, and DHQ production

As above, HHQ, PQS, HQNO, 2-AA, and DHQ production in bacterial cultures was assessed in presence of 11 different concentrations of D88 (50 μM, 25 μM, 12.5 μM, 6.25 μM, 3.125 μM, 1.562 μM, 0.781 μM, 0.390 μM, 0.195 μM, 0.0976 μM, 0.0488 μM). GraphPad Prism 9.2.0 software was used to plot the IC$_{50}$ curve against the percent inhibition of these small molecules' production at each concentration.

## Assessment of MvfR binding to *pqsA* promoter using chromatin immunoprecipitation (ChIP)

The D88 inhibition of MvfR binding to *pqsA* promoter was evaluated using PA14 expressing MvfR fused to a vesicular stomatitis virus glycoprotein (VSV-G) epitope at the C-terminus[22]. Overnight grown culture of PA14 expressing MvfR-VSV-G stain were diluted to an $OD_{600nm}$ 0.01 and grown at 37 °C with and without D88 50 μM and/or PQS (38 μM) and allowed the cells to grow until $OD_{600nm}$ reached 1.0. Thereafter MvfR-DNA complex were cross-linked and isolated via chromatin immunoprecipitation (ChIP). Coprecipitated DNA was purified (using DNA purification kit, Qiagen, USA) and quantified using quantitative real-time polymerase chain reaction qPCR in a Quant-Studio 3 Thermal Cycler (Applied Biosystems, USA). *pqsA* specific oligonucleotides (forward 5'-AAATTTCTCGCGGTTTGGAT-3' and reverse 5'-CAGAACGTTCCCTCTTCAGC-3') were used for quantification, and the percentage of MvfR binding to the promoter was calculated using the input method. Non-MvfR regulated *rpoD* promoter (forward 5'-ACCGTCGTGGCTACAAATTC-3' and reverse 5'-GGCGATCTTCAGTACCTTGC-3') was used as a negative control[22,34].

## Drug Metabolism and Pharmacokinetics (DMPK) of the D88 in mice

Experiments were conducted by Aptuit (Verona) S.rl, *an Evotec Company* in Italy. Six weeks old male CD-1 mice were obtained from Charles River Laboratories, Italy. Mice were maintained on a 12 h light cycle with *ad libitum* access to rodent feed and water.

Healthy animals received IV 1.0 mg kg$^{-1}$ or SC administration of the 10.2 mg kg$^{-1}$ in homogeneous suspension in 0.5 % HPMC in water at the volume of 1.7 mL kg$^{-1}$. Following administration, blood samples were collected under deep isoflurane anesthesia from the cava vein of each mouse into tubes with K3EDTA, thoroughly but gently mixed, and placed on wet ice. Within 0.5 h of collection, blood was centrifuged (2500 g for 10 min at 4 °C) and within 0.5 h, aliquots of plasma were transferred into appropriately labeled sample tubes. A first 20 μL aliquot were mixed with 80 μL of Hepes 0.1 N. A second 5 μL aliquot of plasma was added to a well plate for urea quantification without dilution. Plasma samples were processed using a method based on protein precipitation with acetonitrile followed by HPLC/MS-MS analysis with an optimized analytical method.

## Ethics statement

Animal protocols were reviewed and approved by the Institutional Animal Care and Use Committee (IACUC) at the MGH (protocol no. 2006N000093) and are in strict accordance with the guidelines of the Committee on Animals of the MGH, Harvard Medical School (Boston, USA), and the regulations of the Subcommittee on Research Animal Care of the MGH and the National Institutes of Health. Animals were euthanized according to the guidelines of the Animal Veterinary Medical Association. All efforts were made to minimize suffering. Aptuit (Verona) Srl carried out all experiments involving animals for PK studies, *an Evotec Company*, Italy, in accordance with the European directive 2010/63/UE governing animal welfare and protection, which is acknowledged by the Italian Legislative Decree no 26/2014 and according to the company policy on the care and use of laboratories animals. All the studies were revised by the Animal Welfare Body and approved by the Italian Ministry of Health (authorization n. -PR).

## Infection and D88 treatment studies in mice

Ten-week-old male C57BL/6 mice were purchased from Jackson Laboratories. Mice were maintained in a specific pathogen-free (SPF) environment at the Massachusetts General Hospital (MGH; Boston, USA), in a 12 h light 12 h dark photoperiod at an ambient temperature of 22 ± 1 °C, with food and water access *ad libitum*.

Prior to burn injury, all mice were anesthetized using one 500 μl intraperitoneal (IP) injection of ketamine (125 mg kg$^{-1}$) and xylazine (12.5 mg kg$^{-1}$) in normal saline (N/S), and the dorsal fur was subsequently removed with an electric clipper. A 30% total body surface area (TBSA) dorsal burn was induced by immersion in 90 °C water for 8 sec, using a polystyrene foam template, as in the well-established burn model described by Walker and Mason (1968), with some modifications[69]. Spinal protection from the thermal injury was achieved by a dorsal subcutaneous injection of 500 μL N/S before the induction of the burn injury. Fluid resuscitation was achieved by an intraperitoneal injection of 500 μL N/S.

Immediately after burn injury, 100 μL of 10 mM MgSO$_4$ containing approximately $3 \times 10^5$ colony forming units (CFUs) of *PA* clinical isolate PA14 culture or isogenic *mvfR* mutant culture were intradermally injected at the burn eschar of mice in the burn plus infection (BI) group. Mice in the burn-alone groups received an equivalent injection of 100 μL of 10 mM MgSO$_4$. After the experiment, all animals were returned to their cages to allow recovery from anesthesia. All cages were kept on heating pads during this period to prevent hypothermia. Food and hydrogel on the cage floor were provided *ad libitum*[70].

For the group supplemented with our MvfR-inhibiting compound (D88), mice received five subcutaneous injections (at the nape of the animals) at 1, 4, 8, 12, and 18 h post-BI, at a dose of 24 mg kg$^{-1}$ body weight. D88 was prepared in a 40% Captisol vehicle. Control groups received equivalent doses of 40% Captisol vehicle.

## In vivo intestinal permeability assay

For the assessment of the intestinal barrier function, 4 h before euthanasia, mice were gavaged with 0.2 ml of Fluorescein Isothiocyanate-Dextran (FITC-Dextran; 3–5 kDa; cat. no. FD4; Sigma-Aldrich; Merck KGaA, Darmstadt, Germany) in PBS, so that a dose of 440 mg kg$^{-1}$ body weight was achieved. 22 h post-BI, mice were euthanized. The aseptic cardiac puncture was performed to obtain blood samples. The collected blood was stored in BD microtainer SST amber blood collection tubes on ice and then centrifuged at 15,000 g for 90 seconds. The serum was removed and was used to assess the FITC levels with fluorescent spectrophotometry (excitation, 480 nm, and emission, 520 nm)[70].

## Tissue harvesting

Immediately after euthanasia, ileum and colon samples were aseptically harvested through a midline laparotomy. The intestine samples

were flushed three times with sterile PBS. The samples were either snap-frozen in liquid nitrogen and stored at −80 °C or stored in 4% paraformaldehyde for future analysis.

The entire small and large intestine and the muscle underlying the burn eschar were aseptically harvested in different experiments. The samples were immediately homogenized in 1 mL sterile PBS using a tissue homogenizer (Polytron, PT 10-35), and the homogenate was serially diluted and plated on Pseudomonas-isolation agar plates. Following plating, all plates were incubated at 37 °C, CFUs were quantified after 24–36 h, and the counts were normalized by tissue weight.

### Production of HHQ, PQS, HQNO, 2-AA, and DHQ in vivo

To measure the production of HHQ, PQS, HQNO, 2-AA, and DHQ in vivo, underlying muscle from the site of infection was collected at 12 h and 22 h post-infection, and the small molecules were extracted by homogenizing them immediately in 1 mL sterile PBS as above. A 500 µL of sample was mixed with an equal volume of methanol containing 10 ppm of HHQ-D4 and 20 ppm of PQS-D4, spun down for 5 min, and 700 µL of supernatant were put in glass vials. Liquid chromatography-mass spectrophotometry (LC/MC) analysis was performed as described above and in[66].

### Tight junction (TJ) immunofluorescence assay

Samples of distal ileum fixed with 4% paraformaldehyde were cut in sections and mounted on microscope slides. After deparaffinization and antigen retrieval (Antigen Retrieval reagent; R&D Systems, Minneapolis, MN, USA), tissue sections were immersed in PBS/0.1 tween for 10 min and were blocked by Normal Goat Serum. They were then incubated with primary antibody rabbit polyclonal anti-claudin-1 (Catalog# 71-7800; final concentration, 1:100; Invitrogen, Rockford, IL, USA) overnight in a humid chamber at 4 °C. The sections were washed three times with PBS and secondary antibody goat anti-rabbit (Catalog# ab150061; final concentration: 1:500; Abcam, Cambridge, MA, USA) and DAPI (Catalog# ab228549; Abcam, Cambridge, MA, USA) were added, and incubated 1 h at room temperature. The sections were then washed three times with PBS, dried, and mounted, and images were collected using a confocal microscope (Nikon ECLIPSE Ti2; NIS-Elements AR 5.21.03; Nikon Instruments Inc., Tokyo, Japan).

### Intestinal inflammation assessment

Distal ileal TNF-α and IL-6 were quantified using the mouse TNF-α enzyme-linked immunosorbent assay [ELISA] Ready-SET-Go kit (eBioscience; San Diego, CA, USA) and the Mouse IL-6 DuoSet ELISA (R&D Systems) respectively, as per the manufacturer's instructions.

### Reporting summary

Further information on research design is available in the Nature Research Reporting Summary linked to this article.

## Data availability

The authors declare that data supporting the findings of this study are included in the paper and supporting information except for the Crystal structure of MvfR ligand binding domain in complex with M64 (PDB ID: 6B8A), which is published and available at https://doi.org/10.2210/pdb6b8a/pdb. For all experiments, source data having exact n replicate is provided in a Source Data file. Source data are provided with this paper.

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

## Acknowledgements

This study was supported by the Massachusetts General Hospital Research Scholar Award and grant R01AI134857 to LGR. VKS was in part supported by IUSSTF-SERB Indo-U.S Postdoctoral Fellowship. MA was supported by the Shriners Hospitals Research Fellowship #84313. We thank Dr. Kelsey Wheeler for the critical reading of the manuscript and Dr. Yuji O. Kamatari for the technical advice on molecular docking analysis, and Enamine Ltd., Kyiv, Ukraine for compounds' syntheses. The funders had no role in study design, data collection, interpretation, or the decision to submit the work for publication.

## Author contributions

L.G.R. contributed to the conception of the study. M.A., V.K.S., and L.G.R. contributed to the design of the study. V.K.S., M.A., D.M., and T.K. contributed to data acquisition. V.K.S., M.A., D.M., T.K., and L.G.R. contributed to data analysis and interpretation. F.L., D.A., and L.G.R. contributed to the design of SAR and their analysis. F.S. and M.G. contributed to the synthesis of compounds D42 and D44. A.F., L.F., S.F., G.B., E.C., C.P., M.N., and T.P. contributed to DMPK and SPR studies and interpretation. E.D. and O.P. contributed to Mass Spec data acquisition and analysis. M.A., V.K.S., T.K., and L.G.R. wrote the paper. All authors have approved the submitted version of the study and their contributions.

## Competing interests

L.G.R. has a financial interest in Spero Therapeutics, a company developing therapies to treat bacterial infections. L.G.R.'s financial interests are reviewed and managed by Massachusetts General Hospital and Mass General Brigham Integrated Health Care System in accordance with their conflict-of-interest policies. No funding was received from Spero Therapeutics and had no role in study design, data collection, analysis, interpretation, or the decision to submit the work for publication. The remaining authors declare no competing interests. Patent: Broad Sepctrum anti-virulence anti-persistence compounds. Inventors: Laurence Rahme, Francois Lepine, Damien Maura, Carmella Napolitano, Antonio Felice, Michele Negri, Stefano Fontana, Daniele Andreotti. Institution: Massachusetts General Hospital. Publication number: 20210130306. Filed October 19, 2018. Publication date: May 6, 2021. All compounds reported in this publication are included in the aforementioned patent.

## Additional information

[1]Department of Surgery, Harvard Medical School and Massachusetts General Hospital, Boston, MA 02114, USA. [2]Shriners Hospitals for Children, Boston, MA 02114, USA. [3]Department of Microbiology, Harvard Medical School, Boston, MA 02115, USA. [4]Translational Biology Department, Aptuit (Verona) S.rl, an Evotec Company, 37135 Via A. Fleming 4, Verona, Italy. [5]DMPK Department, Aptuit (Verona) S.rl, an Evotec Company, 37135 Via A. Fleming 4, Verona, Italy. [6]In vitro Chemotherapy Laboratory, Aptuit (Verona) S.r.l., an Evotec Company, 37135 Via A. Fleming 4, Verona, Italy. [7]Centre Armand-Frappier Santé Biotechnologie, Institut National de la Recherche Scientifique (INRS), Laval, Quebec H7V 1B7, Canada. [8]Department of Chemistry, University of Massachusetts Lowell, Lowell, MA 01854, USA. [9]Global Synthetic Chemistry Department, Aptuit (Verona) S.r.l., an Evotec Company, 37135 Via A. Fleming 4, Verona, Italy. [10]Department of Microbiology Discovery, In Vitro Biology, Aptuit (Verona) S.r.l., an Evotec Company, 37135 Via A. Fleming 4, Verona, Italy. [11]Present address: Voyager Therapeutics, Cambridge, MA 02139, USA. [12]Present address: T. Kitao, Department of Microbiology, Graduate School of Medicine, Gifu University, Gifu 501-1194, Japan. [13]Present address: A Felici, Academic Partnership, Evotec SE, 37135 Via A. Fleming 4, Verona, Italy. [14]These authors contributed equally: Vijay K. Singh, Marianna Almpani, Damien Maura. ✉e-mail: rahme@molbio.mgh.harvard.edu

