## [Peer Review File · Nature Communications]

Tackling recalcitrant *Pseudomonas aeruginosa* infections in critical illness via anti-virulence monotherapyREVIEWER COMMENTS

Reviewer #1 (Remarks to the Author):

This is original work from a PI who is known for expertise in this field, especially with *P. aeruginosa* and especially with MvfR. I have very few criticisms of this MS and there is adequate molecular detail to support their conclusions.

The manuscript by Singh et al. "Tackling Recalcitrant *Pseudomonas aeruginosa* Infections In Critical Illness via Anti-virulence Monotherapy" is a logical continuation of an excellent series of publications by the Dr. Rahme's lab on *P. aeruginosa* virulence and the role of MvfR. In the current work, they report novel anti-MvfR compounds that they have tested in vitro and in vivo for their efficacy against this feared pathogen. Overall, the work herein presented high quality, logical, and the conclusions are supported by the data.

The only possible confounder here is that mouse model employed involving a burn wound model of infection is somewhat problematic regarding how *P. aeruginosa* develops in burn patients. Most *P. aeruginosa* that causes sepsis in burn patients is gut derived. Given that the hypothesis tested here is that anti-MvfR agents can provide a strategy to prevent *P. aeruginosa* infections in burn patients, it would have been more logical to use a mouse model of gut-derived sepsis wherein gut colonization of *P. aeruginosa* is created and the sepsis that follows in the burned animal develops as it does clinically. However this is a minor point and does not detract from the excellent findings of this MS.

Minor comments:

The authors used 24 hrs time point to measure biofilm formation (line 254). Usually, the maximal amount of biofilm formation by *P. aeruginosa* occurs at earlier time points, around 6 hrs.

Line 279: correct for "exhibited"

Line 359: correct for "binding"

Line 398: correct for "burned"

Fig. 7A. The panels in PA14+Vehicle are very dark.

There are μ l, μ L, ul (line 686), uL (line 691), mL1-1 (? , line 702), etc. Please make these units uniform.

Reviewer #2 (Remarks to the Author):

The manuscript showed that N-Aryl Malonamides represent a new class of MvfR inhibitors that may be used to control *Pseudomonas aeruginosa* infections. The findings could be interesting should the authors could demonstrate that administration of NAMs, or at least, D88 could significantly reduce the mortality caused by *P. aeruginosa* infection.

Comments and suggestions:

1. Determine whether D88 could increase the rate of mice survival.
2. Provide data on the effect of 10 selected NAMs on bacterial growth.
3. Fig. 6B shows that no significant difference between PA14+D88 and the mvfR mutant, this seems unlikely when comparing the data shown in Fig. 6C.
4. LN187 : Add data in Suppl info and indicate the final concentration tested.

5. LN205: Delete "in the nanomolar range", Fig. S2 used 10 micromolar range.
6. Fig. 6A legend: indicate the symbol blue and red syringes.
7. Change delta MvfR to delta mvfR. delete gene but not protein.

Reviewer #3 (Remarks to the Author):

MvfR is a relatively new appealing target to inhibit biofilm formation and development in *Pseudomonas aeruginosa*, still poorly explored. This strategy is particularly appealing as it blocks bacteria development while avoiding the strong selective pressure associated to antibiotic killing.

The present study represents one of the first global studies targeting this protein in a broader context. Hence it is certainly noteworthy in terms of scope and significance of the results. The study further demonstrates that MvfR promotes intestinal permeability and introduces a new class of confirmed MvfR inhibitors (NAMs) with IC₅₀ in the nanomolar range.

Also, MvfR as a target system within PA is particularly appealing as no clinical isolates from patients have been reported to date to have frequent mutations in Mvfr, contrary to what happens for example in the much more studied LasR.

The motivation is clearly described. A good justification on the SAR studies based on M17 is presented. Several compounds exhibit IC₅₀ at a nanomolar range and dose responsive inhibition. Efficacy of the molecules was tested against 16 isolates. Choice of D88 as the lead compound was based on solubility assessment. Toxicity evaluation was also presented.

Overall, the article is well structured. With some exceptions, the methodology is well documented and follows the standards of the field. Results are analyzed with careful detail. The conclusion are well supported by the results and well anchored in a detailed discussion.

The following minor aspects have been noticed and require correction:

Line109-112. Please add proper reference.

Line 199-201 – Please be more specific about “chemical liabilities” and avoid the use non-specific language such as “appear to bear no chemical liabilities”

Line 365 – please add units to the free energy value presented.

Line 750. Molecular docking. The authors have used the mypresto software, but no reference to this program and its inherent approximations, scoring function used, search algorithm or docking parameters is presented. A general website is presented, but it is impossible to ascertain these details. The accuracy of mypresto in comparison with well-established docking programs is also not presented. Cross-docking results of M64 are also not presented. Hence, this part is no reproducible and impossible to evaluate.

Point-by-Point Response to the Reviewers' Comments

We would like to thank the editor and the reviewers for their time and effort in reviewing our manuscript. Moreover, we are grateful to them for their constructive comments and for acknowledging the high quality and significance of our work and that our data support our conclusions, which were reached with appropriate and rigorous analysis.

Reviewer #1

“This is original work from a PI who is known for expertise in this field, especially with P. aeruginosa and especially with MvfR. I have very few criticisms of this MS, and there is adequate molecular detail to support their conclusions.”

“The manuscript by Singh et al. “Tackling Recalcitrant Pseudomonas aeruginosa Infections In Critical Illness via Anti-virulence Monotherapy” is a logical continuation of an excellent series of publications by the Dr. Rahme’s lab on P. aeruginosa virulence and the role of MvfR. In the current work, they report novel anti-MvfR compounds that they have tested in vitro and in vivo for their efficacy against this feared pathogen. Overall, the work herein presented high quality, logical, and the conclusions are supported by the data.”

We thank the reviewer for the constructive comments and for recognizing our expertise, the importance, and the quality of our work.

The only possible confounder here is that mouse model employed involving a burn wound model of infection is somewhat problematic regarding how P. aeruginosa develops in burn patients. Most P. aeruginosa that causes sepsis in burn patients is gut derived. Given that the hypothesis tested here is that anti-MvfR agents can provide a strategy to prevent P. aeruginosa infections in burn patients, it would have been more logical to use a mouse model of gut-derived sepsis wherein gut colonization of P. aeruginosa is created and the sepsis that follows in the burned animal develops as it does clinically. However this is a minor point and does not detract from the excellent findings of this MS.

We apologize for not clearly conveying the reasoning for selecting the mouse infection model we used to test our compounds. Our goal was not to specifically study infections in burns. The burn wound infection mouse model used serves as a surrogate of the critical illness status and gut permeability and allows to test the systemic consequences of P. aeruginosa infection initiating at a wound site, which is a distant site. Patients admitted to critical care units are at high risk for increased morbidity from the skin and wound infections, many of which are P. aeruginosa-related. We added a better rationale for the choice of the model in the revised text- Line 306-307line

Minor comments:

The authors used 24 hrs time point to measure biofilm formation (line 254). Usually, the maximal amount of biofilm formation by *P. aeruginosa* occurs at earlier time points, around 6 hrs.

Indeed, we used 24h time point as the maximum biofilm formation with strain PA14 grown in a minimal medium (M63) is observed at 24 hours. M63 is the typical medium used in biofilm assessment. Please see the Figure below showing a time course of biofilm formation using a starting PA14 culture at OD_{600nm} 0.01.

Line 279: correct for “exhibited”
Corrected

Line 359: correct for “binding”

Corrected

Line 398: correct for “burned”

Corrected

Fig. 7A. The panels in PA14+Vehicle are very dark.

Correct, the panels of the PA14+vehicle infection group that received no treatment are dark. They reflect the dramatically decreased immunofluorescence staining for claudin-1 compared to the mice infected with PA14 and treated with D88 and animals infected with the isogenic *mvfR* mutant strain. The acquired images are original, and their intensity has been quantified and presented in the supplementary Figure 6.

There are μ l, μ L, ul (line 686), uL (line 691), mLI-1 (? , line 702), etc. Please, make these units uniform.

Corrected

Reviewer #2

The manuscript showed that N-Aryl Malonamides represent a new class of MvfR inhibitors that may be used to control Pseudomonas aeruginosa infections. The findings could be interesting should the authors could demonstrate that administration of NAMs, or at least D88 could significantly reduce the mortality caused by P. aeruginosa infection.

We thank the reviewer for the constructive comments and for recognizing the importance of our work.

Comments and suggestions:

1. Determine whether D88 could increase the rate of mice survival.

This study focuses on intestinal derangement and not on mortality as a result of wound infection in the critically ill setting. Thus, adding survival studies is out of the focus of this manuscript. In a future study, we plan to include survival studies utilizing several additional animal models to test the efficacy of NAMs in acute and persistent infections. Such survival studies will address various routes of infection and compound administration and will have to include all necessary combinations of controls, as we have done in this study.

2. Provide data on the effect of 10 selected NAMs on bacterial growth.

Data are included in the Supplemental Figure 2

3. Fig. 6B shows that no significant difference between PA14+D88 and the mvfR mutant, this seems unlikely when comparing the data shown in Fig. 6C.

If we understand the point of the reviewer correctly, the question is, since there is no significant difference in the FITC-dextran flux between *mvfR* and PA14+D88 (Fig. 6B), which demonstrates the therapeutic efficacy of D88 in restricting the flux of FITC significantly, why the CFUs reported (Fig 6C) from the site of infection are high and the same in all groups?

If this is the question, the CFU results are in agreement with the fact that targeting the MvfR function doesn't impact the growth or viability of *P. aeruginosa* but decreases the ability of the pathogen to be virulent, as is demonstrated by the abolishment of the production of the small molecules at the site of infection (Fig. 6D), the reduction of the systemic *P. aeruginosa* dissemination represented by the CFUs in ileum and colon groups (Fig 6C), and the intestinal derangement shown in Figure 7A.

4. LN187: Add data in Suppl info and indicate the final concentration tested.

The concentration is now shown as Suppl Fig 3A. We have also added growth data in Suppl info shown in the revised Suppl Fig 3B. Instead of including the growth of all 16

strains in the presence of each of the 10 NAMs and thus showing 160 growth curves, we have added representative data generated with 3 of the clinical strains in the presence of 4 individual compounds as part of this Suppl Figure. As we stated in the original version of our manuscript, none of the compounds affected the growth of any of the clinical isolates tested. This is one of the criteriums used to progress the compounds.

5. LN205: Delete "in the nanomolar range", Fig. S2 used 10 micromolar range.

Corrected

6. Fig. 6A legend: indicate the symbol blue and red syringes.

Indicated

7. Change delta MvfR to delta mvfR. delete gene but not protein.

Corrected

Reviewer #3 (Remarks to the Author):

MvfR is a relatively new appealing target to inhibit biofilm formation and development in *Pseudomonas aeruginosa*, still poorly explored. This strategy is particularly appealing as it blocks bacteria development while avoiding the strong selective pressure associated to antibiotic killing.

The present study represents one of the first global studies targeting this protein in a broader context. Hence it is certainly noteworthy in terms of scope and significance of the results. The study further demonstrates that MvfR promotes intestinal permeability and introduces a new class of confirmed MvfR inhibitors (NAMs) with IC50 in the nanomolar range.

Also, MvfR as a target system within PA is particularly appealing as no clinical isolates from patients have been reported to date to have frequent mutations in Mvfr, contrary to what happens for example in the much more studied LasR.

The motivation is clearly described. A good justification on the SAR studies based on M17 is presented. Several compounds exhibit IC50 at a nanomolar range and dose responsive inhibition. Efficacy of the molecules was tested against 16 isolates. Choice of D88 as the lead compound was based on solubility assessment. Toxicity evaluation was also presented.

Overall, the article is well structured. With some exceptions, the methodology is well documented and follows the standards of the field. Results are analyzed with careful detail. The conclusion are well supported by the results and well anchored in a detailed discussion.

We thank the reviewer for the constructive comments and for recognizing the novelty, the significance of the approach in targeting virulence functions, and the quality of our work.

The following minor aspects have been noticed and require correction:

Line 109-112. Please add proper reference.

Three recent references are added

Line 199-201 – Please be more specific about “chemical liabilities” and avoid the use non-specific language such as “appear to bear no chemical liabilities”

We apologize for not using a specific language. We revised the text (Lines 185-188) to read:

Based on all compounds' inhibition profiles in all the functions tested, we selected for advancement the 10 most potent compounds D41, D42, D43, D57, D69, D77, D80, D88, D95, and D100 (Supplementary Table 1-7 and Fig. 2). These compounds do not contain a chemical group or atom such as a sulfur atom or a free amino group that could easily undergo oxidation in vivo.

Line 365 – please add units to the free energy value presented.

We have added the units. Using AutoDock Vina, we now report the binding energy values in the text (line 283-284) and Supplementary Fig. 5 and 6 in the Supporting information.

Line 750. Molecular docking. The authors have used the mypresto software, but no reference to this program and its inherent approximations, scoring function used, search algorithm or docking parameters is presented. A general website is presented, but it is impossible to ascertain these details. The accuracy of mypresto in comparison with well-established docking programs is also not presented. Cross-docking results of M64 are also not presented. Hence, this part is not reproducible and impossible to evaluate.

We apologize for the insufficient description provided on docking. Please see the revised text (Line 637) and Figure 5 describing the reanalysis of docking using a well-known software AutoDock Vina integrated with UCSF Chimera software. Material and Methods section (*In Silico docking of MvfR-D88*) and Supporting Information, including Supplementary Figs S5 and S6, now also show the updated docking analysis results, including the binding energy values. In addition, we ran the cross-docking validation using MvfR and M64 (Fig S5).

REVIEWERS' COMMENTS

Reviewer #2 (Remarks to the Author):

I am satisfied with most of the revisions, except the revised manuscript still lacks sufficient data to show the clinical significance of their findings. Providing the data regarding whether D88 or other NAMs could increase the rate of mice survival would allow readers to assess the clinical potentials of this approach in the practical control of *P. aeruginosa* infection. This is very important as wound infection by the pathogen could be lethal if untreated or treatment failed. Addition of the suggested data could provide clear indication whether these chemicals could merely prevent intestinal derangement and need further optimization, or they are sparkling already. This is agreeable with the aim of this manuscript.

P. aeruginosa infection is hard to control due to its notorious MDR ability. Any substantial progress in this regard would be much welcome, and this is why I read this manuscript with great interest. However, the current presentation seems to leave readers with a feeling of eating half-cooked meals.

Reviewer #3 (Remarks to the Author):

The authors have addressed all the issues previously raised in my previous revision. In my view the paper is now ready for acceptance.

Point-by-Point Response to the Reviewers' Comments

We would like to thank the editor and the reviewers for their time and effort in reviewing our revised manuscript and are grateful for their constructive comments and for acknowledging the high quality and significance of our work.

Reviewer #2

I am satisfied with most of the revisions, except the revised manuscript still lacks sufficient data to show the clinical significance of their findings. Providing the data regarding whether D88 or other NAMs could increase the rate of mice survival would allow readers to assess the clinical potentials of this approach in the practical control of *P. aeruginosa* infection. This is very important as wound infection by the pathogen could be lethal if untreated or treatment failed. Addition of the suggested data could provide clear indication whether these chemicals could merely prevent intestinal derangement and need further optimization, or they are sparking already. This is agreeable with the aim of this manuscript.

P. aeruginosa infection is hard to control due to its notorious MDR ability. Any substantial progress in this regard would be much welcome, and this is why I read this manuscript with great interest. However, the current presentation seems to leave readers with a feeling of eating half-cooked meals.

We appreciate the reviewers comment. Indeed in a future study, we plan to focus on survival studies utilizing several additional animal models to test the efficacy of NAMs in acute and persistent infections. Such survival studies will address various routes of infection and compound administration and will have to include all necessary combinations of controls, as we have done in this study. This study focuses on intestinal derangement and not on mortality as a result of wound infection in the critically ill setting. Thus, adding survival studies is out of the focus of this manuscript.

Reviewer #3

The authors have addressed all the issues previously raised in my previous revision. In my view the paper is now ready for acceptance.